Manuscript prepared for Biogeosciences Discuss.
with version 3.5 of the LATEX class copernicus_discussions.cls.
Date: 27 April 2018

# Climate and marine biogeochemistry during the Holocene from transient model simulations

**Joachim Segschneider**[1], **Birgit Schneider**[1], **and Vyacheslav Khon**[1,2,3]

[1]Institute of Geosciences, Christian-Albrechts University of Kiel, Ludewig-Meyn-Str. 10, D-24118 Kiel, Germany
[2]now at GEOMAR Helmholtz Centre for Ocean Research Kiel, Kiel, Germany
[3]A.M. Obukhov Institute of Atmospheric Physics, Russian Academy of Sciences, Moscow, Russia

Correspondence to: J. Segschneider (joachim.segschneider@ifg.uni-kiel.de)

## Abstract

Climate and marine biogeochemistry changes over the Holocene are investigated based on transient global climate and biogeochemistry model simulations over the last 9,500 yr. The simulations are forced by accelerated and non-accelerated orbital parameters, respectively, and atmospheric $pCO_2$, $CH_4$, and $N_2O$. The analysis focusses on key climatic parameters of relevance to the marine biogeochemistry, and on the physical and biogeochemical processes that drive atmosphere-ocean carbon fluxes and changes of the oxygen minimum zones (OMZs). The simulated global mean ocean temperature is characterised by a mid-Holocene cooling and a late Holocene warming, a common feature among Holocene climate simulations which, however, contradicts a proxy-derived mid-Holocene climate optimum. As the most significant result, and only in the non-accelerated simulation, we find a substantial increase in volume of the OMZ in the Eastern Equatorial Pacific (EEP) continuing into the late Holocene. The concurrent increase of apparent oxygen utilisation (AOU) and age of the water mass within the EEP OMZ can be attributed to a weakening of the deep northward inflow into the Pacific. This results in a large scale mid-to-late Holocene increase of AOU in most of the Pacific and hence the source regions of the EEP OMZ waters. The simulated expansion of the EEP OMZ raises the question if the deoxygenation that has been observed over the last five decades could be a - perhaps accelerated - continuation of an orbitally driven decline in oxygen. Changes in global mean biological production and export of detritus remain on the order of 10%, with generally lower values in the mid-Holocene. The simulated atmosphere-ocean $CO_2$-flux would result in similar-magnitude atmospheric $pCO_2$ changes as observed for the Holocene, but with different timing. More technically, as the increase in EEP OMZ-volume can only be simulated with the non-accelerated model simulation non-accelerated model simulations are required for an analysis of the marine biogeochemistry in the Holocene. Notably, also the long control experiment displays similar magnitude variability as the transient experiment for some parameters. This indicates that also long control runs are required when investigating Holocene climate and marine biogeochemistry, and that some of the Holocene variations could be attributed to internal variability of the atmosphere-ocean system.

# 1 Introduction

Numerical models that combine the ocean circulation and marine biogeochemistry have been developed since the 1980s (e.g., Maier-Reimer et al., 1993; Maier-Reimer, 1993; Six and Maier-Reimer, 1996; Maier-Reimer et al., 2005). Few studies of marine carbon cycle variability during the Holocene have been performed, however, as the focus of marine carbon cycle research has been more on recent and future climate change related carbon cycle changes (e.g., Maier-Reimer and Hasselmann, 1987; Maier-Reimer et al., 1996), and glacial-interglacial changes (e.g., Heinze et al., 1991; Brovkin et al., 2016; Bopp et al., 2017). However, one thousand year long transient climate experiments have been performed for the last millennium with comprehensive Earth system models that include the marine carbon cycle, (e.g., Jungclaus et al., 2010; Brovkin et al., 2010), and more recently the CMIP5/PMIP3 Millennium experiments (Atwood et al., 2016; Lehner et al., 2015).

Of the many features that characterise the biogeochemical system in the ocean, here we will concentrate on oxygen minimum zones (OMZs), atmosphere-ocean carbon fluxes, and the marine ecosystem. OMZs have received particular attention in the recent past. This is in large part due to the observation that in the last five decades, a general deoxygenation of the world's ocean, and an intensification of the ocean's main OMZs has occurred (e.g., Stramma et al., 2008; Karstensen et al., 2008; Schmidtko et al., 2017). A further decrease of oceanic $O_2$ concentrations has been projected for the future with numerical models (e.g., Matear and Hirst, 2003; Cocco et al., 2013; Bopp et al., 2013) as a consequence of anthropogenic climate change. Knowing the past variations of the OMZ extent and oxygen is, therefore, of immediate relevance to estimate the importance of the observed and projected deoxygenation (e.g., Bopp et al., 2017).

A few studies that investigate past oxygen variations have already been performed: Based on a model study with an intermediate complexity model to investigate glacial-interglacial variations of oxygen, Schmittner et al. (2007) found a causal relationship of Indian and Pacific ocean oxygen abundance and a shut down of the Atlantic Meridional Overturning Circulation (AMOC). In their experiments, AMOC variability was generated by freshwater perturbations.

An attempt to better understand the currently observed and future projected expansion of the OMZ based on paleoceangraphic observations (Moffitt et al., 2015) indicates an expansion of the major OMZs in the world ocean concurrent with the warming since the last deglaciation (18-11 kyr BP, kilo years before present). This is based on estimates of seafloor deoxygenation using snapshots at 18, 13, 10, and 4 kyr BP. Bopp et al. (2017) investigated oxygen variability from the last glacial maximum (LGM) into the future based on CMIP5 simulations (PiControl, the historical and the future period) and time slice simulations of the last LGM (21 kyr BP) and the mid-Holocene (6 kyr BP).

Although the focus of this manuscript is on marine biogeochemistry, it is mainly the changes in climate that are driving the changes in marine biogeochemistry. Hence, some characteristics of the Holocene climate variability need to be addressed. Model-based investigations of Holocene climate are performed under the auspices of the Paleo Model Intercomparison Project (PMIP, Braconnot et al., 2012). Initially, numerical model time slice experiments have been used to simulate the climate at specific time intervals, typically 9.5 kyr, 6 kyr, and 0 kyr BP. The simulated climate and its variability has been compared to proxy data (e.g., Leduc et al., 2010; Emile-Geay et al., 2016). Also transient experiments over the entire Holocene have been performed, mainly with accelerated orbital forcing to save computing time (Lorenz and Lohmann, 2004; Varma et al., 2012; Jin et al., 2014) or coupled atmosphere-ocean intermediate complexity models with non-accelerated forcing (Renssen et al., 2005, 2009; Blaschek et al., 2015). Longer model simulations exist also for Earth system models of intermediate complexity (EMICS), such as described in Brovkin et al. (2016) for the last 8 kyr, and for 6 kyr BP to 0 kyr BP with a comprehensive ocean-atmosphere-land biosphere model but orbital forcing only (Fischer and Jungclaus, 2011). The longest non-accelerated climate simulation with a comprehensive model is the Tra-CE 21ka model experiment with the Climate Community System Model 3 for the last 21 kyr (Liu et al., 2014).

A second source of information about climate variability during the Holocene comes from proxy data. A concerted effort to synthesise these estimates by the PAGES2K project has resulted in a temperature reconstruction over the last 2,000 years in fairly high temporal resolution (PAGES 2k Consortium, 2013). In this reconstruction, the global mean surface air temperature

is analysed to cool by about 0.3°C between 1000 A.D. and 1900 A.D, followed by a sharp increase in temperature. Before 1000 A.D. the temperature is fairly constant at about 0.1 °C colder than the 1961-1990 average.

Wanner et al. (2008) also provide a comprehensive overview of globally collected proxy-based climate evolution for the last 6,000 yr together with some instructive plots of the insolation changes during that period based on Laskar et al. (2004). For land-based proxies the authors consistently find a decrease of temperatures from 6 kyr BP until now, with different amplitude, but for the ocean this is more heterogenous (Wanner et al., 2008, Fig. 2). E.g., the sea surface temperature (SST) displays an increase with time in the subtropical Atlantic, whereas SST decreases in line with the land surface records in the western Pacific and in the North Atlantic (see also Marchal et al., 2002).

A continuous reconstruction of temperatures for the entire Holocene, i.e., the past 11,300 years, albeit with lower temporal resolution before the PAGES2K period has been assembled by Marcott et al. (2013). In their reconstruction, global mean surface air temperature increases by about 0.6 °C between 11.3 kyr BP and 9 kyr BP to 0.4 °C warmer than present (as defined by the 1961-1990 CE mean). After 6 kyr BP temperatures slowly decrease by 0.4 °C until 2 kyr BP and are relatively stable for 1,000 years. This is followed by a relatively fast decrease beginning around 1 kyr BP of 0.3 °C, in agreement with the PAGES2K data and an increase to present day temperatures in the last few hundred years before present (Marcott et al., 2013, Fig. 1a-f).

Model simulations and proxy-based estimates of past climate variability apparently show some disagreement (Liu et al., 2014), and the model simulations described here make no exception. One reason may be a different behaviour of land and ocean, as e.g., the PMIP2 model simulations shown in Wanner et al. (2008) show warmer mid-Holocene temperatures over land, in particular over Eurasia, whereas there is little SST difference between 6 kyr BP and modern values. Also on shorter timescales there are descrepencies between model results and proxy-based records. E.g., proxy-based estimates indicate changing El Niño-Southern Oscillation (ENSO) related variability during the Holocene that cannot be reproduced by most of the PMIP models (Emile-Geay et al., 2016). Also the proxy-derived inverse relationship between ENSO

variability and the amplitude of the seasonal cycle is not picked up by most of the models (Emile-Geay et al., 2016, Fig. 3). The reasons for the mismatch in proxy-based and model-simulated Holocene climate variability, despite some efforts in the PMIP community, have yet to be established.

In this manuscript we aim at closing the gap between glacial-interglacial and future greenhouse gas (GHG) driven simulations of climate and the marine carbon cycle and earlier time-slice experiments of the Holocene. Given the differences in simulated and proxy-derived climate evolution over the Holocene, this study should be regarded as a sensitivty study to orbital and GHG forcing. Following earlier time slice experiments with a coupled atmosphere-ocean-sea-ice climate model and a marine biogeochemistry model (Xu et al., 2015), here we are using transient model simulations with a comprehensive model system that are covering the last 9.5 kyr of the Holocene. In particular, we investigate the temporal evolution of some of the key elements of the simulated climate that are important drivers of marine biogeochemistry variations, such as SST and AMOC. For the marine carbon cycle we focus on global values of primary production, export production, and calcite export, all of which can result in atmosphere-ocean $CO_2$-flux and OMZ variations. Based on these results we analyse and discuss changes in the OMZs, in particular in the EEP but also in the Atlantic and the Arabian Sea, the integrated effect of changes in the atmosphere-ocean $CO_2$-flux, and changes in the marine ecosystem.

In addition we want to address the more technical question to what extent simulations with accelerated orbital forcing are suitable for Holocene marine biogeochemistry simulations. In the accelerated-forcing experiments, the change in orbital parameters between two model years corresponds to a 10 yr step in the real orbital forcing (see Sec. 2.2.1). For climate simulations, the sensitivity to accelerated vs. non-accelerated forcing has recently been investigated for the last two interglacials (130-120 kyr BP and 9-2 kyr BP, Varma et al., 2016), indicating that non-accelerated experiments differ from accelerated experiments in the representation of Holocene climate variability in the higher latitudes of both hemispheres while the behaviour is more similar in low latitudes. A different temporal evolution was also found for the deep ocean temperature, with the non-accelerated experiment displaying a larger variation. Here we perform and analyse simulations of the marine biogeochemistry of the Holocene forced by an

accelerated and a non-accelerated climate model simulation of the Holocene.

We will first describe the numerical models, the experiment setup, and characteristics of the time-varying forcing in Sec. 2, report the results for climatic and biogeochemical variables in Sec. 3, and discuss the results and implications for future research in Sec. 4.

## 2 Model description and experiment set-up

### 2.1 Models

#### 2.1.1 The Kiel Climate Model (KCM)

Oceanic physical conditions are obtained from the global coupled atmosphere-ocean-sea-ice model KCM (the Kiel Climate Model, Park and Latif, 2008; Park et al., 2009), in particular from NEMO/OPA9 (Madec, 2008), which comprises the oceanic component of KCM and includes the LIM2 sea-ice model (Fichefet and Morales Maqueda, 1997). The atmospheric component is ECHAM5 (Roeckner et al., 2003). The spatial configuration for ECHAM5 is T31L19, and for NEMO the ORCA2 configuration is chosen, i.e., a tripolar grid with a nominal resolution of $2°$x $2°$and a meridional refinement to $0.5°$near the equator and 31 layers with a finer resolution in the upper water column. The upper 100 m are resolved by 10 layers, and below the euphotic zone there are 20 layers with increasing thickness up to a maximum of 500 m for the deepest layer.

KCM has previously been used to conduct and analyse time-slice simulations of the pre-industrial and the mid-Holocene climate and hydrological cycle (Schneider et al., 2010; Khon et al., 2010, 2012; Salau et al., 2012) and contributed to PMIP3 (e.g., Emile-Geay et al., 2016). More recently, orbital forcing (eccentricity, obliquity, and precession) were varied continiously over the last 9,500 yr of the Holocene according to the standard protocol of PMIP (Braconnot et al., 2008). This forcing was accelerated by a factor of 10, resulting in a transient model experiment of 950 model years for the Holocene (Jin et al., 2014). Here, in additional KCM experiments, the forcing is non-accelerated, so that the Holocene is represented by 9,500 model

years starting from 9.5 kyr BP (see Sec. 2.2.1 for the experiment description).

### 2.1.2 Pelagic Interactions Scheme for Carbon and Ecosystem Studies (PISCES)

Monthly mean fields of temperature, salinity, and the velocity from the KCM experiment were used in off-line mode to force a global model of the marine biogeochemistry (PISCES, Aumont et al., 2003).

Since the description of PISCES in Aumont et al. (2003) is quite comprehensive, we restrict the model description to the most relevant parts for our investigation. Sources of oceanic oxygen are gas exchange with the atmosphere at the surface, and biological production in the euphotic zone. Oxygen consuming heterotrophic aerobic remineralisation of dissolved organic carbon (DOC) and particulate organic carbon (POC) is simulated over the whole water column, i.e., also in the euphotic layer. Remineralisation depends on local temperature and $O_2$-concentration. For an increase of 10 °C the rate increases by a factor of 1.8 ($Q_{10}$=1.8). Remineralisation is reduced for $O_2$-concentrations below 6 $\mu$mol l$^{-1}$.

Primary production is simulated by two phytoplankton groups representing nanophytoplankton and diatoms. Growth rates are based on temperature, the availability of light and the nutrients P, N (both as nitrate and ammonium), Si (for diatoms), and the micronutrient Fe. The elemental ratios of iron, chlorophyll, and silicate within diatoms are computed prognostically based on the surrounding water's concentration of nutrients. Otherwise they are constant following the Redfield ratios. Photosynthetically available radiation (PAR) is computed from the shortwave radiation passed from ECHAM to NEMO. Sea ice is assumed to reflect all incoming radiation so there is no biological production in areas that are completely sea-ice covered (i.e., where the sea-ice fraction is equal to 1).

There are three non-living components of organic carbon in PISCES: semi-labile DOC, as well as large and small POC, which are fuelled by mortality, aggregation, fecal pellet production and grazing. In the standard version of PISCES, large and small POC sinks to the sea floor with their respective settling velocities of 2 and 50 m d$^{-1}$. For large POC, the settling velocity increases further with depth. In the model version employed here, the simulation of the settling velocity of large detritus is formulated allowing for the ballast effect of calcite and opal shells

according to Gehlen et al. (2006). The settling velocity of small POC remains constant at 2 m $d^{-1}$. In most areas and at most depths, the ballast parametrization leads to a reduction of the settling velocity for large POC compared to the (50 m $d^{-1}$ and more) standard version. The new formulation of the settling velocity for large POC generally improved the oxygen fields of the KCM-driven PISCES simulation when compared to modern day WOA data (Garcia et al., 2013), in particular in the EEP (see Appendix A for a comparison of observed and simulated oxygen distribution and a sensitivity of the OMZ-volume to the $O_2$-threshold). Note that the ballast parameterization was not part of the PISCES version used in Xu et al. (2015), and therefore the mean state of the EEP OMZ differs between the experiments of Xu et al. (2015) and the ones described here.

We also added an age tracer to PISCES. The age tracer is set to zero at surface grid points, and then the age increases with model time elsewhere. Advection and mixing is also applied to the age tracer.

## 2.2 Experiment setup

### 2.2.1 KCM - greenhouse gases and astronomical forcing

As GHG and orbital forcing are the boundary conditions driving the forced variations in the KCM experiments, we describe this forcing in a little more detail. We do not take into account changes in total solar irradiance (TSI), sea level, changes in ice sheets (neither topography nor albedo), fresh water input into the North Atlantic, or volcanic aerosols.

Greenhouse gas concentrations were obtained from the PMIP data base (https://www.paleo.bristol.ac.uk/~ggdjl/pmip/pmip_hol_lig_gases.txt) based on ice cores from the EPICA site (Augustin et al., 2004) and are displayed in Fig. 1a). Prescribed atmospheric $CO_2$ concentration varied from 263.7 ppm at the beginning of the Holocene, decreased to 260 ppm around 7 kyr BP in the mid-Holocene, and then increased to about 274 ppm for the present day pre-industrial conditions (based on Indermühle et al., 1999). $CH_4$ varied from 678.8 ppb in the early Holocene to 580 ppb around 5 kyr BP, slowly increasing afterwards to 650 ppb around 0.5 kyr BP, followed by a steeper increase to 805 ppb during the last five hundred years of the Holocene

reflecting early land use change. $N_2O$ variations were smaller, from 260.6 ppb in the early Holocene to 267 ppb around 2.5 kyr BP.

Eccentricity remained fairly constant at a value of 0.02 over the entire Holocene. The precessional index increased from -0.015 to 0.02, and the obliquity decreased from about 24.2°to 23.5°. In general, this leads to less insolation during northern hemisphere summer, and more insolation in southern hemisphere summer: Solar radiation at the top of the atmosphere (TOA) in June decreased during the Holocene from 9.5 kyr BP to 0 kyr BP by about 25 $Wm^{-2}$ at the equator, and 45 $Wm^{-2}$ at 60°N. On the southern hemisphere, the decrease is up to 10 $Wm^{-2}$ at 30°S, and at 60°S there is a weak increase of a few $Wm^{-2}$. In December the insolation is stronger for 0 kyr BP than for 9.5 kyr BP by up to 30 $Wm^{-2}$ at 30°S and about 5 $Wm^{-2}$ at 60°N (see also Jin et al. (2014, Fig. 2) and Wanner et al. (2008, Fig. 6) for changes in solar radiation at top of atmosphere vs. time for different latitudes and summer/winter, based on Berger and Loutre (1991) and Laskar et al. (2004), respectively).

We note that the total annual radiation driven by precession changes remains fairly constant at each latitude and globally, whereas obliquity changes cause changes also in the annual mean insolation (see e.g., Fig.1b in Schneider et al., 2010). These annual mean changes in TOA insolation from 9.5 kyr BP to 0 kyr BP are an increase of around 5 $Wm^{-2}$ at the poles and a decrease of 1 $Wm^{-2}$ at the equator, thereby potentially decreasing the latitudinal temperature gradient.

For our analyses that focus on ocean physical conditions and marine biogeochemistry, however, we need to consider the TOA forcing as filtered by the atmosphere, i.e., at the sea surface. In Fig. 1b annual and zonal mean anomalies of short wave radiation (SWR) at the ocean and sea-ice surface are displayed as a Hovmöller diagramme. These annual mean anomalies are somewhat different from the TOA anomalies, but more relevant to understand the simulated SST evolution and changes in PAR. In the early Holocene, negative anomalies of -1 to -3 $Wm^{-2}$ develop at high latitudes of mainly the southern hemisphere. In the mid-Holocene negative anomalies of -1 to -3 $Wm^{-2}$ start to evolve also in northern high latitudes, and are -3 to -5 $Wm^{-2}$ in the southern high latitudes around 60°S, whereas SWR-anomalies become positive in low latitudes (1 to 3 $Wm^{-2}$). At around 60°N, there is a shift from negative to positive

anomalies at around 6.8 kyr BP. During the late Holocene, the positive anomalies in the low latitudes intensify (3 to 5 $Wm^{-2}$), whereas the high latitude anomalies remain about constant.

### 2.2.2 KCM experiments (KCM-CTL, KCM-HOLx10 and KCM-HOL)

The basis for the KCM experiments is a 1,000 year KCM experiment with 9.5 kyr BP orbital parameters, 286.6 ppm $CO_2$, 805 ppb $CH_4$, and 276 ppb $N_2O$ concentration (with a final global average SST of 15.8°C), followed by a spin up for a further 1,000 years with 9.5 kyr BP orbital and 9.5 kyr BP GHG forcing ($pCO_2$=263.8 ppm, $CH_4$=678.8ppb and $N_2O$=260.6 ppb). From this state the KCM-CTL and KCM-HOL experiments were started. The KCM control experiment (KCM-CTL) was integrated for a further 7,860 years with orbital parameters and atmospheric greenhouse gases kept constant at 9.5 kyr BP values as continuation of the spinup experiment. Due to computational limitations, it was not possible to run KCM-CTL for the full 9,500 yr. In the transient experiments KCM-HOLx10 (950 years) and KCM-HOL (9,500 years), time varying orbital parameters and greenhouse gases as described in Sec. 2.2.1 were applied as forcing.

### 2.2.3 Spinup of PISCES and control experiment (BGC-CTL)

To spin up the biogeochemical model, monthly mean ocean model output from experiment KCM-CTL was used as forcing. This then available 2,000 yr long forcing (first 2,000 years of KCM-CTL) was repeated three times to spin-up PISCES for 6,000 years, after which period the model drift as defined by air sea carbon flux and age of water masses was negligible. It was in particular the age tracer in the deep northern Pacific that required the long spinup time. Note that this BGC spin-up simulation does not achieve a 'classical' time-invariant steady state but reflects the internal variability of the first 2,000 years from experiment KCM-CTL and any remaining drift. After repeating the KCM-CTL forcing three times for the spinup, PISCES was integrated for a further 7,860 years with the available KCM-CTL forcing as a control experiment for the marine biogeochemistry (BGC-CTL).

Discussion Paper | Discussion Paper | Discussion Paper | Discussion Paper | Discussion Paper |

### 2.2.4 Transient experiments with PISCES (BGC-HOLx10,BGC-HOL)

Similarly to the set-up of the Holocene KCM experiments, we performed two transient experiments with PISCES in off-line mode. Both transient experiments are also started from year 6,000 of the PISCES spinup experiment. In the accelerated experiment BGC-HOLx10, oceanic fields of KCM-HOLx10, and the same atmospheric $pCO_2$ as in KCM-HOLx10 is prescribed as forcing. In this experiment, PISCES is integrated for 950 years corresponding to the period 9.5 kyr BP to 0 kyr, with 10-fold accelerated forcing. Monthly mean output is stored. The non-accelerated experiment BGC-HOL is integrated for 9,500 years forced by the non-accelerated experiment KCM-HOL and the corresponding $pCO_2$. All experiments and their names are summarised in Tab. 1.

Note that the approach here differs from earlier work to investigate Holocene OMZ changes with a KCM/PISCES model setup, where PISCES was forced by PMIP-protocol time-averaged oceanic conditions for specific time slices (6 kyr BP and 0 kyr BP, Xu et al., 2015). Also, now all BGC experiments make use of the direct KCM-NEMO output, as opposed to the setup in Xu et al. (2015) where KCM-derived anomalies were added to mean ocean fields from an reanalysis-forced ocean-only setup.

### 2.3 Processing of model output

All plots in the results section are based on model output interpolated to a regular $1°$x $1°$grid using the CDO/SCRIP interpolation package. The only exception is the meridional overturning circulation (MOC) that has been computed on the original ORCA2 grid for different ocean basins using the standard cdf-tool available from the NEMO-package (https://github.com/meom-group/CDFTOOLS). Maximum values have been computed using the Ferret @max function.

For all time series the time-axis represents the forcing years. This corresponds to model years for the non-accelerated experiments but not for the accelerated experiments, so any variation caused by long term internal variability of the model would be spread out in time in the accelerated experiment compared to the non-accelerated experiment. For all time series, the

long term changes are indicated by the 4th-order polynomial fits from the xmgrace software package. An exception is alkalinity, where polynomial fits are of 8th-order to allow for the higher curvature of the time series. Dots represent annual averages and their spread indicates interannual to centennial time scale variability. Plots for BGC-HOL and BGC-CTL are based on output from every 10th year, both to be consistant with BGC-HOLx10, and to keep the output file size at a managable level.

## 3 Results

### 3.1 Climate variations over the Holocene

Since the biogeochemical variations depend to a large extent on the changes in ocean physics, we will first examine the relevant aspects of the simulated climate variations over the Holocene.

### 3.1.1 Sea surface temperature

As a first indicator of simulated changes in ocean physics, we present time series of the global and annual mean SST (Fig. 2a). The global mean SST in KCM-HOL is 15.1 °C at 9.5 kyr BP, decreases to 14.8 °C at 6.5 kyr BP, and increases to 15.6 °C at 0 kyr BP (based on 4th-order polynomial fits). In KCM-HOLx10, the temporal evolution is similar as in KCM-HOL, but with a smaller decrease of global mean SST in the early Holocene (to 15.0 °C and a slightly higher SST than KCM-HOL at the end of the late Holocene (15.7 °C).

Also the control experiment KCM-CTL displays a decrease of global mean SST of about 0.1 °C per 1,000 years over its first 3,500 yr integration time. This drift is reducing to 0.1 °C per 2,000 years between 6 kyr and 4 kyr BP, and after 4 kyr BP the drift becomes very small. Note that the drift over the first 500 yr is negligible in KCM-CTL (grey bar in Fig. 2a). That even a five hundred year period of stable SST does not guarantee a later drift-free climate in the control simulation is rather unexpected.

The SST evolution in KCM-CTL implies that the simulated early Holocene decrease in SST in KCM-HOL and KCM-HOLx10 is the combined result of a remaining model drift, and the

orbital and $CO_2$ forcing. The initial SST decrease would be weaker, and the SST increase from mid-to-late Holocene would be stronger in a drift-free experiment KCM-HOL.

A Hovmöller diagramme of zonal mean SST anomalies of KCM-HOL (Fig. 2b) reveals that the mid-Holocene cooling is strongest in the higher latitudes of the southern hemisphere (up to -0.75 °C, centered at around 60°S) whereas the late Holocene warming is strongest between 40°S and 40°N with maxima around the equator and at 40°S. This pattern coincides to large extent with that of the anomalies of SWR at the ocean and sea-ice surface (Fig. 1b).

The seasonal cycle of global mean SST in KCM-HOL doubles its amplitude from around 0.35 °C in the early Holocene to 0.7 °C at about 3 kyr BP (Fig. A.3a), indicating the dominance of the increasing seasonal cycle in the solar forcing on the southern hemisphere mid-latitudes over the decreasing seasonal cycle on the northern hemisphere (Jin et al., 2014). The seasonal cycle of global mean SST remains in the range of slightly less than 0.7 °C during the late Holocene after 3 kyr BP.

### 3.1.2 Meridional overturning circulation

The Atlantic meridional overturning circulation (AMOC) serves as an indicator of the intensity of deep water formation in the source region of the global conveyor belt. From the NEMO-package output, maximum AMOC at 30°N has been computed (Fig. 3). Based on the 4th-order polynomial fits shown in Fig. 3, the simulated maximum AMOC at 30°N in KCM-HOL at 9.5 kyr BP is around 13.9 Sv. AMOC is gradually decreasing to slightly more than 12.5 Sv until 3 kyr BP, indicating a weak slow down of the global conveyor belt circulation. AMOC then marginally increases until the end of the Holocene to around 12.6 Sv (Fig. 3).

In KCM-HOLx10 the mean AMOC and its temporal evolution are similar to KCM-HOL, with a slightly higher mean value. The control experiment KCM-CTL, however, also displays changes in AMOC, similar to the changes in KCM-HOL. Overall, the long term changes in AMOC are relatively small in all experiments and remain within the range of interannual to centennial variations of around 2-3 Sv.

In the Pacific, the deep northward flow that forms the far end of the deep branch of the conveyor belt circulation also weakens with time during the Holocene. Between 3000 and 5000

m depth, at latitude 0°N, the decrease of the maximum flow is from almost 10 Sv at 9.5 kyr BP to 7.5 Sv at 0 kyr BP (dashed line in Fig. 3), indicating a reduced replenishment with younger waters in the deep Pacific. Also in the Indian Ocean the deep inflow from the South is slightly decreasing with time but less strongly (not shown).

### 3.1.3 Age of water masses

In addition to AMOC, the age of water masses can serve as an indicator of deep water formation, the intensity of the global deep water circulation, and help to understand changes in oxygen concentration. We will investigate time series of the water mass age in the deep ocean at the source and end regions of the global convyeor belt circulation, namely the North Atlantic and the North Pacific.

The renewal of water masses in the North Atlantic is indicated by a time series of the age tracer averaged between 1,800 m and 2,500 m depth and 40°W to 10°W, 40°N to 60°N in Fig. 4a. The average water mass age in this volume in BGC-HOL initially ranges from 60 to 80 yr over the first 2,800 yr, followed by a sudden decrease to slightly more than 25 yr that occurs within a few years around 6.8 kyr BP. This is followed by a gradual increase to around 40 yr over the remaining 6,800 yr of the Holocene. The sudden decrease is likely driven by changes in SST in the North Atlantic which in turn are a consequence of the changing solar radiation in this area. We will come back to this point in Sec. 4.3.

Also the control experiment BGC-CTL, however, simulates a sudden decrease in water mass age similar to the one in BGC-HOL but occuring at a different time. In the accelerated experiment BGC-HOLx10 (brown curve in Fig. 4a) a slightly weaker decrease in age from 60 to 30 yr is simulated for the deep North Atlantic, but it occurs over a longer time period (roughly 300 model years) and later in terms of forcing years (between 4 kyr and 1 kyr BP).

At the far end of the conveyor belt circulation, the deep North Pacific, changes occur less sudden than in the North Atlantic, but with a larger amplitude. Between 2,500 m and 3,500 m depth, 150°E to 130°W, 40°N to 60°N the water masses show an initial age of 1,475 yr for all experiments (Fig. 5). In BGC-HOL, water mass age initially decreases to around 1,400 yr around 7.5 kyr BP, but from thereon there is a steady increase up to an age of 1,800 yr at the

Discussion Paper | Discussion Paper | Discussion Paper | Discussion Paper | Discussion Paper |

end of the Holocene. Also in BGC-CTL the water mass age in the deep North Pacific increases after 8.5 kyr BP, but less strong than in KCM-HOL. At 1.6 kyr BP, the end of BGC-CTL, the water mass age is 1,650 yr compared to nearly 1,800 yr in KCM-HOL.

This can not be simulated in the accelerated experiment BGC-HOLx10, however, that runs for 950 yr only. In BGC-HOLx10 deep North Pacific water mass age decreases slightly stronger than in the control experiment to 1,400 yr at 0 k BP. The increase in water mass age in the non-accelerated experiment BGC-HOL is indicating a considerable slow down of the global conveyor belt circulation over the Holocene with significant impact on the marine biogeochemistry in the Pacific. We will come back to the age of water masses when investigating the evolution of the EEP OMZ in section 3.2.5.

## 3.2 Biogeochemical variations

### 3.2.1 Ocean-Atmosphere carbon flux

In this section the atmosphere-ocean carbon flux is diagnosed. As atmospheric $pCO_2$ is prescribed in all BGC experiments, the diagnosed flux is a combination of the climate driven oceanic variations, and the prescribed $pCO_2$. We will come back to this point in the discussion (Sec. 4.5).

In the early Holocene the atmosphere-ocean carbon flux in BGC-HOL is around -0.5 GtC $yr^{-1}$ (Fig. 6a), the equilibrium value in the PISCES model, indicating an outgassing that is balancing riverine carbon input. In the mid-Holocene the carbon flux is slightly reduced to around -0.4 GtC $yr^{-1}$. Indicating slightly stronger outgassing, the value increases to -0.75 GtC $yr^{-1}$ in the late Holocene in experiment BGC-HOL, whereas the flux remains at around -0.4 GtC $yr^{-}1$ in experiment BGC-HOLx10. In BGC-CTL, the $CO_2$ flux varies between -0.45 and -0.6 GtC $yr^{-1}$. The amplitude of the seasonal cycle of the atmosphere-ocean carbon flux in BGC-HOL decreases from early to late Holocene from around 1.8 GtC $yr^{-1}$ to only 0.8 GtC $yr^{-1}$ (Fig. A.3b).

The time-integrated atmosphere-ocean carbon flux n BGC-HOL (blue curve in Fig. 6a) is almost zero during the early Holocene, and increases to 10 GtC from 7 kyr BP to 4.5 kyr BP.

From 4 kyr BP to 0 kyr BP there is a steady decrease to -42 GtC, indicating a net flux from the ocean to the atmosphere in the late Holocene.

The zonal mean changes of the atmosphere-ocean carbon flux in BGC-HOL (Fig. 6b) are indicating a change from net $CO_2$ uptake to outgassing in the high latitude Southern Ocean and mostly increased uptake in northern mid-latitudes. Also the positive anomaly around 40°S shows stronger uptake from mid-to-late Holocene.

### 3.2.2 Surface alkalinity and pH

In BGC-HOL total alkalinity (TA) at the sea surface increases from 2240 to 2250 $\mu$mol l$^{-1}$ until 8 kyr BP and increases further to 2265 $\mu$mol l$^{-1}$ between 6.5 and 4 kyr BP (Fig. 7a). After 4 kyr BP, TA decreases to 2258 $\mu$mol l$^{-1}$ in the late Holocene. In experiment BGC-HOLx10 the global mean concentration of TA remains in the range of 2240 to 2245 $\mu$mol l$^{-1}$, with a maximum at around 6 kyr BP. Surprisingly, also in BGC-CTL TA increases considerably and even slightly stronger than in BGC-HOL.

The increase in TA in BGC-HOL occurs over most latitudes (Fig. 7b) with a stronger increase north of 40°N and around the equator between 5 kyr BP and 3 kyr BP, whereas there is only a small trend around 60°S. This temporal evolution can only partly be explained by a reduction in CaCO$_3$ export that would drive an increase in TA (Sec. 3.2.4, Fig. 11b).

The global and annual mean pH at the surface is following the temporal variations in atmospheric pCO$_2$ and varies only little during the Holocene, with changes in the range of a few hundredths pH-units (8.13 - 8.16, not shown).

### 3.2.3 Nutrients

In BGC-HOL, the global mean NO$_3$ concentration averaged over the euphotic zone (0-100 m) is decreasing with time from 56 $\mu$mol l$^{-1}$ in the early Holocene to 52 $\mu$mol l$^{-1}$ around 3 kyr BP. This is followed by a slight increase to almost 53 $\mu$mol l$^{-1}$ at 0 kyr BP (Fig. 8a). In BGC-HOLx10, the global mean concentration is fairly constant at 56 $\mu$mol l$^{-1}$ until 5 kyr BP, and then declines gradually to 54 $\mu$mol l$^{-1}$ in the late Holocene. In BGC-CTL, the decrease in the

global mean $NO_3$ concentration is similar to that in BGC-HOL until 7 kyr BP, and becomes weaker thereafter. However, $NO_3$ concentration is not increasing after 3 kyr BP as it does in BGC-HOL.

The Hovmöller diagramme of the zonal mean $NO_3$ concentration changes of experiment BGC-HOL (Fig. 8b) reveals that the decrease of the $NO_3$ concentration originates from a large range of latitudes mainly in the southern hemisphere (40°S to 60°S), and also from high northern latitudes after the 'event' around 6.8 kyr BP in the North Atlantic centered at 60°N. The weak increase of global mean euphotic-zone $NO_3$ concentration after 3 kyr BP originates mainly from a small band centered at 55°N, counteracted by a weakening of the negative anomalies around and south of the equator (10°N to 20°S).

### 3.2.4   Marine ecosystem

Here the focus is on the three major components of the marine ecosystem with relevance for the carbon cycle, namely the integrated primary production, the export production, and the calcite (calcium carbonate) export. The primary production integrated over the euphotic zone (INTPP) is a measure of the productivity of the marine ecosystem. INTPP in BGC-HOL is around 44 GtC yr$^{-1}$ at the beginning of the Holocene, decreasing to a minimum of around 41 GtC yr$^{-1}$ in the mid-Holocene at 5 kyr BP, and then increasing again to 44 GtC yr$^{-1}$ towards the late Holocene (Fig. 9a). Interannual variations are about 2-3 GtC yr$^{-1}$. In the accelerated experiment BGC-HOLx10, INTPP remains fairly constant over the entire Holocene at 43 to 44 GtC yr$^{-1}$. In the control experiment BGC-CTL INTPP decreases steadily form 44 GtC yr$^{-1}$ to aroud 41 GtC yr$^{-1}$ at the end of the simulation.

The decrease in global mean INTPP in BGC-HOL originates mainly from latitudes south of 40°N and is generally more pronounced on the southern hemisphere (Fig. 9b). The increase after 4 kyr BP can be traced back to an increase in INTPP between 40°N and 60°N beginning around 6 kyr BP and intensifying and gradually spreading southward for the remainder of the Holocene. This response is likely driven by a combination of the changes in SST, PAR, and nutrient availability (Figs. 2b, 1b, 8b), as there is some similarity between zonal mean anomalies of INTPP and SST, SWR at the sea/sea-ice surface, and $NO_3$, but none of the patterns is matched

excactly.

The export production at 100 m depth in BGC-HOL, here computed as sum of small and large POC (see Sec. 2.1.2), is around 10.2 GtC yr$^{-1}$ at 9.5 kyr BP. During the early and mid-Holocene, there is a slight decrease to 9.8 GtC yr$^{-1}$ at 4 kyr BP, and export production remains fairly constant at that level until 3 kyr BP after which there is a modest increase to 10 GtC yr$^{-1}$ (Fig. 10a). The accelerated experiment BGC-HOLx10, after a small increase in the early Holocene, simulates a relatively uniform decrease by just 0.1 GtC yr$^{-1}$ from 8 kyr BP to 0 kyr BP. Also the control experiment BGC-CTL simulates a quite uniform decrease of export production, from 10.2 GtC yr$^{-1}$ at the beginning to 9.9 GtC yr$^{-1}$ at the end of the experiment.

The zonal mean export production in BGC-HOL decreases mainly in the low latidues in two bands centred around 20°N and 35 °S (Fig. 10b). An increase occurs mainly between 30°N and 60°N, intensifying after 6.8 kyr BP. The apparent deviations from the pattern of INTPP could be explained by changing temperatures (with an impact on the remineralisation rate) and changes in the particle composition (with an impact on settling velocity) and relative contributions from small and large POC to the export production. For slowly sinking small POC, we find a more continuous but minor decline of global export from 3.7 to 3.6 GtC yr$^{-1}$, whereas for the faster sinking large POC the decline is more rapid during the first 3,000 yr of the Holocene (from 6.5 to 6.3 GtC yr$^{-1}$) and the export is fairly constant thereafter (not shown).

The temporal evolution of the calcite export in all experiments is similar to that of INTPP: In BGC-HOL, calcite export is around 1.08 GtC yr$^{-1}$ in the early Holocene (Fig. 11a), followed by a decrease of about 0.1 GtC yr$^{-1}$ (10%) until the mid-Holocene (around 6 kyr BP) after which there is a slight increase again to 1.05 GtC yr$^{-1}$ towards the late Holocene. In BGC-HOLx10 the calcite export fluctuates fairly constantly around its initial value of about 1.08 GtC yr$^{-1}$ whereas in BGC-CTL an initially stronger decline from 1.075 GtC yr$^{-1}$ to 1 GtC yr$^{-1}$ at 1.6 kyr BP is simulated.

The zonal mean changes of CaCO$_3$ export in BGC-HOL are similar to those of INTPP, with an almost global decrease in the early Holocene, and a recovery in the higher northern latitudes after 6.8 kyr BP. The recovery gradually extends to the entire northern hemisphere in the late Holocene (Fig. 11b).

Overall the variations of the global marine biological production and export rates remain in the range of +/-10% throughout the Holocene even in the non-accelerated experiment BGC-HOL, with a tendency for lower values in the mid-Holocene, and surprisingly similar-magnitude variations in the control run.

### 3.2.5 Oxygen minimum zones

The largest OMZ in the global ocean resides in the EEP. The EEP here is defined as the region from 140°W - 74°W, 10°S-10°N, and to compute the EEP OMZ-volume a threshold of 30 $\mu$mol l$^{-1}$ is used. The volume of the EEP OMZ initially remains fairly constant in BGC-HOL from 9.5 kyr BP to 7 kyr BP at $15 \times 10^{14} m^3$ (Fig. 12, left y-axis, dashed lines). But from 7 kyr BP onwards, the OMZ-volume steadily increases to around $26 \times 10^{14} m^3$ at 0 kyr BP in KCM-HOL, an increase of more than 70%. In the accelerated forcing experiment BGC-HOLx10 the EEP OMZ-volume remains fairly constant over the entire Holocene. Also in BGC-CTL the EEP OMZ-volume increases after 8 kyr BP, but less strong than in BGC-HOL.

At the same time as the OMZ-volume increases in BGC-HOL, the age of the water mass within the OMZ increases from around 440 yr (9.5 - 7 kyr BP) to 530 yr at 0 kyr BP (Fig. 12, right y-axis, solid lines). Note that the accelerated experiment does not show an increase in the OMZ-volume and water mass age (Fig. 12). In BGC-HOLx10 the age decreases mainly after 6 kyr BP from 430 to 415 yr. The longer control run, however, also shows substantial variations of OMZ-volume and water mass age. In BGC-CTL the water mass age increases to 490 yr at the end of the experiment with a stronger increase in the last 1,500 years of the experiment.

Time series of the oxygen saturation (O$_2$sat) and apparent oxygen utilisation (AOU) between 100 and 800 m depth in the EEP for BGC-HOL demonstrate a relatively stable O$_2$sat (following mainly the temperature evolution) but an increase in AOU from 252 to 267 $\mu$mol l$^{-1}$ (Fig. 13a). Late Holocene minus early Holocene differences show that O$_2$sat is decreasing in the upper 400 m of the EEP by up to 5 $\mu$mol l$^{-1}$, but increasing by up to 4 $\mu$mol l$^{-1}$ below that depth (Fig. 13b, shading). The corresponding temperature change is a warming of up to 0.4 °C in the 100 - 400 m depth range, and a cooling of up to 0.4 °C below 400 m depth, with an overall very similar pattern as for AOU (Fig. 13b, contours).

Discussion Paper | Discussion Paper | Discussion Paper | Discussion Paper |

For AOU, there is a more uniform-with-depth tendency to higher values in the late Holocene, with AOU up to 25 $\mu$mol l$^{-1}$ higher at around 1000 m depth and slightly lower late Holocene AOU values only near the surface (Fig. 13c, shading). The corresponding change in idealised water mass age ranges from 10 to 60 years between 400 m and 1000 m, with a similar pattern to the AOU changes (Fig. 13c, contours).

Export production in the EEP is fairly constant over the Holocene at 0.58 GtC a$^{-1}$(figure not shown), with only a marginal tendency for lower values in the late Holocene, and thus can be ruled out as a driver of the expansion of the EEP OMZ. The average O$_2$ concentration within the OMZ decreases slightly from 18.5 $\mu$mol l$^{-1}$ at 9.5 kyr BP to 17.5 $\mu$mol l$^{-1}$ at 0 kyr BP in BGC-HOL (not shown).

In contrast to the EEP, for the OMZ in the tropical Atlantic mainly south of the equator, the changes over the Holocene are more modest and of opposite sign. In the region from 5°W - 15°E, 30°S - 5°N, the volume of the OMZ in BGC-HOL decreases slowly from around 4 x $10^{14}$m$^3$ to 3.5 x $10^{14}$m$^3$, and the average age over the OMZ decreases from about 125 yr to 115 yr (Fig. 14).

For the Arabian Sea a steady increase in OMZ-volume is simulated in BGC-HOL (1 x $10^{14}m^3$ to 6 x $10^{14}m^3$), concurrent with an increase in water mass age from 100 to 120 yr (Fig. 15). In the accelerated experiment BGC-HOLx10, there is a similar increase of both OMZ-volume whereas mean water mass age increases mainly before 6 kyr BP. In BGC-CTL average water mass age and OMZ-volume are much less variable than for the transient experiments. Note that the results for the Arabian Sea in Gaye et al. (2017) are from an earlier accelerated experiment, started at year 1,500 of KCM-CTL. The earlier results are quite similar but not identical to BGC-HOLx10.

## 4 Discussion

### 4.1 Holocene SST variations

Comparing the KCM-simulated temporal evolution of global mean SST with observation-based estimates and other model simulations, there is a notable difference between models and observations. During the proxy-derived climate optimum in the mid-Holocene (8 kyr to 5 kyr BP) observation-based global mean temperature is about 0.4 °C warmer than 1961-1990 (Marcott et al., 2013), and borehole temperatures from Greenland are about 2 °C warmer (Dahl-Jensen et al., 1998). During the same period the KCM-simulated SST is at its lowest value, about 0.8 °C colder than at the end of the simulation at 0 kyr BP in the non-accelerated experiment (Fig. 2a).

The largest fraction of the initial post-glacial temperature increase in the reconstructions of Marcott et al. (2013), however, occurs in the very early Holocene (11.3 kyr BP to 9 kyr BP), whereas the simulations discussed here start at 9.5 kyr BP to avoid difficulties with the simulation of retreating ice masses and increasing sea level. Simulations, therefore, start at a time when continental ice sheets and sea level are assumed to be close to present day values. This very early Holocene temperature increase can, therefore, not be simulated by KCM in its present configuration. We note that also in the Holocene time slice experiments with KCM, the annual mean SST is lowest for the 6 kyr BP experiment, and highest for the 0 kyr BP experiment (e.g., Schneider et al., 2010, Fig.6).

In support to our model results, and raising the general question of how representative the mainly land based proxy-derived temperature anomalies can be transfered to the Holocene SST, Varma et al. (2016) find a similar temporal evolution of global mean SST in their orbitally forced Commuity Climate System Model version3 simulations of the Last and Present Interglacials (LIG and PIG, respectively) mainly in their non-accelerated PIG experiment (their Fig.4). The larger amplitude of our simulated temperature change can be explained by the small cooling trend still inherent in the control run (KCM-CTL, about 0.1 °C/ 1,000 years, an otherwise very acceptable value) and the additional forcing from the transient $CO_2$ and $CH_4$ variations in KCM-HOL and KCM-HOLx10 with a range of about 20ppm for $CO_2$, and of about 100 ppb for

CH$_4$ (Fig. 1a), whereas Varma et al. (2016) used constant pCO$_2$ values during their simulations.

From earlier experiments with KCM with/without a 1%/2% p.a. atmospheric pCH$_4$ increase a contribution on the order of 0.1°C to the simulated Holocene global mean SST variation from the prescribed CH$_4$ seems a reasonably conservative estimate (Biastoch et al., 2011, supplementary material Fig. S3). We also performed an additional transient non-accelerated KCM experiment with constant pCO$_2$ of 286.4 ppm but orbital forcing for the Holocene that has not been discussed here. In that experiment, the global mean SST fluctuates within a constant range almost until 4 kyr BP, i.e., with no mid-Holocene cooling, and increases by 0.2 to 0.3 °C from 4 kyr BP to 0 kyr BP. This indicates that the early to mid-Holocene SST evolution in KCM-HOL is a result of the GHG forcing and any remaining model drift, whereas for the late Holocene both GHG and orbital forcing drive an increase in SST.

We note that also in the simulations of Varma et al. (2016) seemingly small variations in atmospheric pCO$_2$ lead to larger variations in the simulated global mean SST than expected from climate sensitivity estimates: For their PIG and LIG simulations with atmospheric pCO$_2$ of 280 ppm and 272 ppm, respectively, the initial global mean SST difference is more than 0.35 °C. This sensitivity is of similar magnitude as the 0.6 °C for KCM control simulations with a pCO$_2$ of 286.4 ppm and 263 ppm.

As such, the Holocene simulations of Varma et al. (2016) and the LGM-to-present simulations discussed in Liu et al. (2014) support the results of our simulations as technically sound. The discrepancy between simulated SST, and proxy-based estimates, however, raises the question of why the simulations result in such a different temporal evolution than observation-based temperature reconstructions.

Possibly there might be a difference in the behaviour of the SST and the mainly land-based temperature reconstructions. E.g., Renssen et al. (2009) display simulated differences between 9 kyr BP and 0 kyr BP, and their Fig. 3c suggests mainly colder temperatures of the northern hemisphere oceans for 9 kyr BP, whereas the trend is opposite for the land surface mainly in Eurasia. Leduc et al. (2010) investigate Mg/Ca ratios and alkenone unsaturation values (U$^{K'}_{37}$) for a range of sediment cores from different locations. In their study, the majority of the U$^{K'}_{37}$ records suggests a Holocene warming trend in the EEP and the tropical Atlantic, whereas Mg/Ca

records indicate a Holocene cooling trend in the same regions. For the North Atlantic, the alkenone-derived SST decreases over the Holocene, but Mg/Ca-derived SST shows a differing warming/cooling trend for the various records. Apparently, further research is needed on this issue.

The model/data mismatch could also imply that at least early Holocene temperature variations were determined not only by orbital forcing or greenhouse gases but also by solar and volcanic forcing, ice sheets, and internal variability of the system (see also Wanner et al., 2008; Renssen et al., 2009, for a more complete and regional investigation of driving mechanisms of Holocene climate). However, including total solar irradiance (TSI) in the forcing would likely not solve the problem. Reconstructed TSI variations over the Holocene are around 1 W m$^{-2}$ (Vieira et al., 2011), and assuming a climate sensitivity of 0.5 K (W m$^{-2}$)$^{-1}$ (IPCC, 2007) these could translate into global mean temperature variations of similar magnitude as simulated here, but reconstructed TSI is relatively low during the mid-Holocene.

As proxies might be seasonally biased (e.g., Schneider et al., 2010), we also analysed the northern summer (June-July-August, JJA) and northern winter (December-January-February, DJF) SST separately, but did not find a better match with the proxy-derived mid-Holocene warming. Also using the yearly maximum temperature, indicating local summer, does not change the temporal behaviour, it only shifts the SST curve upward by roughly 2 K. In summary, the simulated large scale SST evolution in KCM-HOL is seemingly not very sensitive to the choice of season. The Holocene climate conundrum (Liu et al., 2014) is still not solved.

## 4.2 Holocene circulation changes

The meridional overturning in the Atlantic in KCM-HOL decreases over the Holocene by about 1.5 Sv/10%, whereas the deep inflow into the Pacific decreases by 2.5 Sv/20% (Sec. 3.1.2, Fig. 3). As the slow down of the circulation in the deep Pacific in experiment KCM-HOL seems stronger than one would expect from the decrease of the AMOC this suggests a decoupling of the North Atlantic and the North Pacific over the course of the Holocene simulations. This indicates that changes in the 'far-end' conveyor belt circulation are not necessarily represented by changes in AMOC.

Blaschek et al. (2015) describe a set of Holocene experiments including various forcings with the earth system model of intermediate complexity "LOVECLIM" and compare their results with available proxies for AMOC, including those investigated by Hoogakker et al. (2011), see Table 2 in Blaschek et al. (2015). Since the temporal evolution of AMOC in our experiment KCM-HOL is similar to that of experiment 'OG' (Orbital and Greenhouse gases as forcing) in Blaschek et al. (2015), we can assume that their findings are also valid for our experiments: Additional forcing with the 8.2 kyr BP fresh water pulse and also ice sheet topography changes seems to be required to simulate the weak early Holocene AMOC derived from proxies. As those forcings are not included in our model experiment, there is a further reason to discuss our experiments as a sensitivity experiment to orbital and GHG forcing. Also Renssen et al. (2005) find in their experiments with a coupled intermediate complexity model that AMOC remains fairly constant throughout their 9,000 yr Holocene experiment, despite the changes in location of the deep water formation regions.

## 4.3 North Atlantic

Our original intention in examining the North Atlantic more closely was to investigate whether the changes in the OMZs could be traced back to the deep water source regions. It turned out, however, that significant changes occurred in the North Atlantic, that justify further analysis. In section 3.1.3 we showed a sudden drop in the water mass age in the deep North Atlantic (Fig. 4a) that can be traced back to a westward shift in the deep water formation areas south of Iceland, and a northward shift north of Iceland, as indicated by the difference in the annual maximum of the mixed layer depth in the North Atlantic (Fig. 4b). In addition to the shift of location, also an increase of the mixed layer depth of up to 3000 m occurs in the more southwestern part of the Nordic Seas. This is accompanied by a change in SST at 60°N around 6.8 kyr BP from negative to positive anomalies (Fig. 2b). and an increase in export production (Fig. 10b).

An SST time series at 53°N, 30°W shows a rapid decrease in SST, a time series at 62°N 30°W a slight SST increase, and both time series have reduced variability after 7 kyr BP (figure not shown). Fig. 1b reveals a shift from negative anomalies of annual mean SWR at the ocean/sea-ice surface just north of 60°N to positive anomalies in a narrow band south of 60°N,

and a negative anomaly north of 75°N. In particular the positive anomaly south of 60°N seems surprising, as TOA radiation in the North Atlantic is decreasing from 9.5 kyr BP to 7 kyr BP from around 190 W m$^{-2}$ to 180 W m$^{-2}$. This decrease is more pronounced in summer, with a potential for preconditioning the water masses for winter convection, whereas winter insolation is very low anyway so any changes would be of limited impact. The TOA SWR northern hemisphere decrease is fairly linear and continues further for the entire Holocene. The abrupt response of KCM to the forcing anomalies suggests that there may be a critical threshold for a sudden shift of deep water formation areas in the (model) North Atlantic during the Holocene.

This shift is also visible in the concentration of NO$_3$ (Fig. 8b) and the marine ecosytem as indicated by INTPP, and export production that both increase south of 60°N after 6.8 kyr BP and decrease north of 65°N (Figs. 9b, 10b).

As also the control simulation shows a sudden shift in deep North Atlantic water mass age, this indicates that small variations in the range of the internal (model) variability are sufficient to trigger shifts in the pattern of North Atlantic deep mixing.

## 4.4 Holocene OMZ variations

The dominant mechanisms for past and future OMZ-variability have yet to be established (Jaccard and Galbraith, 2012; Bopp et al., 2017). Potential processes are changes in export production, setting the amount of detritus that can be remineralised, temperature changes that effect oxygen solubility and organic matter remineralisation, and changes in circulation. Circulation changes are changing the time during which oxygen can be consumed in subsurface waters due to remineralisation of organic matter, and the rates at which deoxygenated water masses can be replenished.

E.g., Deutsch et al. (2011) analyse an ocean general circulation model forced by atmospheric reanalysis from 1959-2005 to identify the main mechanism for oxygen mimimum zone variability in the more recent past. From the analyses the authors find that downward shifts of the thermocline and hence an uplifting of the Martin curve (Martin et al., 1987), drive an increase of the suboxic volume due to higher respiration rates at the same water depth. Accordingly, upward shifts of the thermocline cause decreasing suboxic volume.

On glacial to interglacial time scales, on/off changes of the AMOC have been identified to drive OMZ variations also in the Pacific (Schmittner et al., 2007). In their model experiments with the UVic (University of Victoria) intermediate complexity model, however, the AMOC collapses entirely, whereas the AMOC in our Holocene simulations varies only by around 10%. As such we find no indication for a direct connection between AMOC changes and the EEP OMZ for more modest changes in AMOC.

Bopp et al. (2017) suggest a compensation of temperature driven changes in $O_2$-saturation and ventilation driven changes in AOU to explain past and future $O_2$ trends. In addition to the variations in idealised water mass age, that can serve as an indicator for ventilation, we computed also the fields of $O_2$-saturation and AOU for experiment BGC-HOL. While $O_2$-saturation in the EEP has a similar temporal evolution as SST, AOU reflects the general slow-down of the circulation in the Pacific (see Fig. 13a). Meridional sections in the EEP reveal a different Holocene-trend of $O_2$-saturation in the upper and lower part of the OMZ: a decrease in the upper part, and an increase below 400 m, reflecting the temporal evolution of temperature. AOU resembles water mass age changes, and is stronger in the late Holocene at all depths with the exception of a small decrease in the upper 100 m off the equator.

Thus, the long term changes show a decoupling of oxygen saturation and AOU, the former driven by temperature changes, the latter by a slower circulation (increase in water mass age). We note that this differs from the compensation of $O_2$-saturation and AOU changes shown in Bopp et al. (2017) who investigate shorter simulations of much stronger climate variations, such as for present day to future, LGM to mid-Holocene, and the shorter time-scale fluctuations from the piControl simulations. In these simulations, a decrease in $O_2$-saturation has a signature in lower AOU and vice versa. However, also Bopp et al. (2017, Fig. 4b) find a correlation between AOU and idealised water mass age for the LGM to mid-Holocene changes, and question if the compensating effect holds also for longer multi-century simulations. The results shown here indicate that for millennial time scales the $O_2$-saturation and AOU changes may be decoupled.

In summary, the dominating mechanism for OMZ changes in our transient, non-accelerated Holocene experiment is a slowdown of the circulation that develops over thousands of years. This slow down is not mainly from changes in AMOC, but is more confined to the deep Pacific.

It results in widespread oxygen consumption and an increase in AOU with an effect on the EEP OMZ (Fig. 12) where the deep waters are upwelled. In the Atlantic, circulation changes remain small over the Holocene, and the OMZ-volume remains fairly constant in the Atlantic OMZ (Fig. 14).

In contrast to studies based on global warming and $O_2$ (Cocco et al., 2013; Bopp et al., 2013) and marine biological production (Steinacher et al., 2010), that showed an impact of stratification changes on $O_2$ fields and marine biological production, here we simulate much smaller temperature variations. The global and annual mean of the MLD in KCM-HOL reveals little temporal variability: MLD is around 48 m at 9.5 kyr BP, and starts to decrease after 5.5 kyr BP to around 47 m at 0 kyr BP. We, therefore, state that stratification changes play only a minor role for large scale $O_2$ changes during the Holocene.

A modest deoxygenation of the global ocean has been observed over the last 50 years (-2% globally, Schmidtko et al., 2017). Moreover, a further decline of the oxygen content, and hence an extension of the world's OMZs has been projected in ocean and Earth system model simulations of the next century as a consequence of global warming (Matear and Hirst, 2003; Cocco et al., 2013; Bopp et al., 2013). As the non-accelerated experiment BGC-HOL yields an expanding OMZ in the EEP during the late Holocene, and in general declining oxygen concentrations as a result of the natural forcing only, this raises the question whether the presently observed deoxygenation might also partly be caused by the continuation of an externally forced trend. This would not have to be in contradiction with an anthropogenic contribution to the observed oxygen decline, but could also point to an amplifaction of the externally forced trend by the effects of anthropogenic warming as both mechanisms point in the same direction, albeit with different time-scales.

Comparing our results with the proxy-derived estimates of OMZ intensity in the Eastern Tropical South Pacific (Salvatteci et al., 2016), we find an indication of stronger denitrification towards the late Holocene in the proxy data. This would likely support our result of an expanding EEP OMZ, but the proxies could also have recorded a more local decrease in $O_2$-concentration, rather than a general expansion of the OMZ. Moreover, the proxy-data for the early Holocene show a decrease in $\delta^{15}N$, indicating increasing oxygen concentrations, which

is, however, not simulated by our model.

An earlier version of the accelerated experiment BGC-HOLx10 was compared to sediment core based estimates of Holocene OMZ evolution in the Arabian Sea. Based on $\delta^{15}N$ records, Gaye et al. (2017) find that the AS OMZ has intensified since the last LGM, and that most of this increase occured throughout the Holocene (their Fig. 6), which is in agreement with the model results presented here. In summary, however, as we do not simulate nitrogen isotopes (or other proxies), a direct comparison between proxies and model results is somewhat limited. A more comprehensive comparison of proxy data and our model results has been planned for the future.

## 4.5 Atmosphere-ocean $CO_2$ fluxes

Concerning the observed changes in atmospheric $pCO_2$ over the Holocene, Indermühle et al. (1999) postulate that changes in the terrestrial biosphere and SST were driving the observed changes. Elsig et al. (2009) base their investigation on $\delta^{13}C$ measurements from an Antarctic ice core. They attribute the 5 ppm early Holocene decrease in atmospheric $pCO_2$ to an uptake of the land biosphere, and the mid-to-late Holocene increase of 20 ppm to changes in the oceans carbonate chemistry. Coral reef formation on newly flooded shelves (Vecsei and Berger, 2004) has been identified as a possible source of atmospheric $CO_2$ in the early Holocene. A comprehensive EMIC-based investigation of interglacial carbon cycle dynamics and potential mechanisms that drive atmospheric $CO_2$ changes during these periods can be found in Brovkin et al. (2016).

Here we can also use our model results to some limited extent to investigate for our model system if and how the ocean may have contributed to the observed atmospheric $pCO_2$ variations. Limitations arise from that the BGC experiments are forced with observed $pCO_2$, and don't include coral reefs nor calcium carbonate compensation from sediments. Potential mechanisms in our model come from the three 'carbon pumps' of the ocean, i.e. changes of oceanic temperature and circulation (physical pump), alkalinity (alkalinity counter pump), and biological production and export of organic matter resulting in changes of the strength (Six and Maier-Reimer, 1996; Ducklow et al., 2001; Segschneider and Bendtsen, 2013) and the efficiency (Sarmiento and

Gruber, 2006) of the biological pump.

During the early Holocene the integrated carbon flux is constant (Fig. 6a), despite a decrease in atmospheric $CO_2$ (Fig. 1a): During this initial phase, the global mean SST in KCM-HOL is decreasing by about 0.3°C (Fig. 2), and the alkalinity in BGC-HOL is increasing by 10 $\mu$mol/l (Fig. 7a), while export production is slightly decreasing by 0.2 GtC yr$^{-1}$ (Fig. 10a). In summary, these effects balance out during the early Holocene, and the time-integrated atmosphere-ocean carbon flux remains about zero until 7 kyr BP (Fig. 6a, blue curve).

In the mid-Holocene, from 7 kyr BP to 4 kyr BP, the time-integrated carbon flux is slowly increasing to a total ocean uptake of 10 GtC. This occurs during a period of atmospheric $pCO_2$ increase that would drive oceanic uptake, but SST is rising slowly, export production is further decreasing, and alkalinity is increasing by a further 10 $\mu$mol/l.

In the late Holocene, after 4 kyr BP, the ocean is outgassing a total of 50 GtC, potentially driven by the simulated increase in SST and damped by the increasing prescribed atmospheric $pCO_2$.

The general slowdown of the circulation over the Holocene suggests a weakening of the physical pump, and a strengthened biological pump as more dissolved inorganic carbon (DIC) from remineralisation of organic matter is stored in the deep ocean. Finally, as a result of reduced calcite export, simulated global mean surface alkalinity increases mainly during the mid-Holocene (6.5 kyr to 5 kyr BP), which would lead to decreasing $pCO_2$ in surface waters and hence a sink for carbon in the atmosphere.

To investigate whether the efficiency of the biological pump ($E_{BP}$, Sarmiento and Gruber, 2006, eq. 4.1.1) changed over the Holocene we computed $E_{BP}$ from the global mean $NO_3$ concentrations from 0 - 100 m depth (surface) and 100 - 200 m depth (deep), as suggested by Sarmiento and Gruber (2006, Fig.4.1.7). It was found that there is little variation of $E_{BP}$ over the Holocene, with values for $E_{BP}$ of around 0.47 +/- 0.01 and a marginal tendency for increased efficiency in the late Holocene (an increase from 0.468 to 0.473 based on a 4th-order polynomial fit), i.e., opposite to the simulated atmosphere-ocean flux that indicates stronger outgassing in the late Holocene.

In summary, the contribution of oceanic processes to air-sea carbon fluxes is in line with the

prescribed $pCO_2$ in the mid-Holocene, while reversing the flux expected from the atmospheric forcing in the early and late Holocene.

## 4.6   Non-accelerated, accelerated, and time-slice experiments

Finally we discuss the gain from the transient experiments performed here compared to the earlier time-slice climate model experiments at 9.5 kyr BP, 6 kyr BP, and 0 kyr BP with KCM (Schneider et al., 2010) and and also the biogeochemistry experiments with PISCES (Xu et al., 2015). Note that the experiments in Schneider et al. (2010) were also performed with preindustrial $pCO_2$ (286.4 ppm) and hence neglect the forcing from changing atmospheric $pCO_2$ that is included here.

One obvious gain from the transient experiments is the more complete time coverage over the Holocene, potentially allowing better comparison with proxies and providing more continuous information. The transient experiments can also be used to determine if the timing of the time-slice experiments is appropriate. For the physical fields like global mean SST (Fig. 2a) we find that the time-slice experiments fit with the extrema of the transient experiment well. As the state of the biogeochemistry, but also of AMOC in the early and late Holocene are not much different, time slice experiments of these periods would miss any changes in between.

Also the time-slice experiments including a 6 kyr BP simulation do not capture the extrema of the simulated time series of the transient experiments BGC-HOL and BGC-HOLx10. E.g., the integrated primary production and export of detritus have their lowest values at 5 kyr to 4 kyr BP (Fig. 9a, 10a). In particular, the evolution of the EEP OMZ in the non-accelerated experiment BGC-HOL was not captured in the earlier time slice experiments of Xu et al. (2015), even though they also showed smaller mid-Holocene OMZs caused by changes in the equatorial current system.

So, are non-accelerated experiments generally required, or can the same knowledge be obtained from experiments with accelerated astronomical forcing? This has already been investigated for physical models (Varma et al., 2016) where a good agreement for physical variables has been found between accelerated and non-accelerated transient simulations over the Holocene in low latitudes, but deviations were found in high latitudes and the deep ocean.

When comparing our results for the accelerated and non-accelerated experiments, we find that also the global mean SST shows some deviations, but it is mainly for the biogeochemical system (that was not included in the Varma et al. (2016) model) that large deviations are simulated. This is mainly the case for globally averaged fields. More regionally, in the EEP OMZ there is a large discrepancy between accelerated and non-accelerated experiment OMZ-volume, whereas results a similar for the Arabian Sea OMZ.

## 5   Conclusions

In this study, a 9,500 year simulation of Holocene climate and marine biogeochemistry is analysed together with a 10-fold accelerated simulation and - to our knowledge for the first time - a control run of similar length as the non-accelerated experiment. The simulated climate in terms of global mean SST is characterised by a mid-Holocene cooling, and a late Holocene warming following the temporal evolution of the greenhouse gas forcing and the short wave radiation at the sea surface. This is in contradiction to a proxy-derived mid-Holocene climate optimum and a late Holocene cooling. The open question why KCM and other ocean-atmoshere coupled climate models do not simulate the proxy-derived Holocene climate evolution under astronomical and GHG-forcing remains a major issue. As long as this issue is not resolved, we have to regard the biogeochemistry simulation as a sensitivity study to orbital and greenhouse gas forcing of the climate system.

Most of the characteristic variables of the marine carbon cycle, like global atmosphere-ocean $CO_2$-flux, primary and export production, and global mean surface pH display changes of up to 10% during the Holocene in the non-accelerated experiment. Variations are generally smaller in the accelerated experiment, but for some variables also the control experiment displays similar magnitude variations as the non-accelerated experiment. This - surprisingly large - variablility in the control experiment is a combination of a small remaining model drift but mainly the internal variability of the climate and marine biogeochemisty system. This internal variability is of similar magnitude as the - orbitally and greenhouse gas - forced variablility during the Holocene. An exception is the EEP OMZ-volume that increases by more than 50% in the

second half of the Holocene in the non-accelerated experiment, but not in the accelerated experiment and much weaker in the control run. This increase in OMZ-volume can be attributed to a weakening deep northward inflow into the Pacific, and a corresponding wide-spread increase in water mass age and AOU that develops over thousands of years. In summary, the simulations demonstrate the necessity of performing transient simulations with non-accelerated forcing when examining the marine biogeochemistry changes over the Holocene, but also of control runs of similar length.

## Appendix A    Simulated and observed oxygen concentrations

Fig. A.1 shows simulated and observation-based (Garcia et al., 2013) $O_2$-concentration profiles in the three major oxygen minimum zones (OMZ) in the world ocean. For all areas, the simulated oxygen concentrations at the surface are overestimated due to the cold bias in KCM-simulated SST compared to present day estimates of observed SST (Locarini et al., 2013). The near-surface oxygen gradient (upper 200 m) is simulated well in the eastern equatorial Pacific (EEP) and the Arabian Sea (AS), but overestimated in the tropical South Atlantic (SATL). Between 200 and 1000 m, the observed concentrations are matched well in the SATL, and differ not too much in the EEP, whereas the observations are poorly matched in the AS, mainly due to a lack of faster sinking large detritus in this area. In general, all BGC simulations show a tendency for too high $O_2$ concentrations in the upper water column, and too low concentrations below 1000 m, with the exception of the AS. Overall, the representation of oxygen minimum zones is within the range of large scale global ocean biogeochemistry models, that all have their errors (Cabré et al., 2015).

As the threshold for the definition of an OMZ is not very well defined in the literature, we display the OMZ-volume for a range of threshold values for observations and as simulated in Fig. A.2 for a) the global ocean and b) the EEP. For thresholds up to 70 $\mu$mol l$^{-1}$, the simulated OMZ-volume is generally too low for the upper 1000 m, while the best match is at a threshold of 80 $\mu$mol l$^{-1}$ for the global ocean and at 70 $\mu$mol l$^{-1}$ for the EEP 0-5000 m range. For higher thresholds, the model simulations overestimate the OMZ-volume.

## Appendix B    Changes in the seasonal cycle

Fig. A.3 demonstrates the increasing amplitude of the seasonal cycle of SST and the decreasing amplitude of the seasonal cycle for the atmosphere-ocean carbon flux during the Holocene. For SST, the annual range increases during the first half of the Holocene from about 0.35 °C to about 0.7 °C at 5 kyr BP and remains at that level for the remainder of the Holocene.

For the atmosphere-ocean carbon flux, the seasonal cycle is almost 2 GtC yr$^{-1}$ in the early Holocene. It becomes continously weaker as the Holocene proceeds and is less than 1 GtC yr$^{-1}$ in the late Holocene. Until 5 kyr BP it is mainly the maximum outgassing that becomes weaker, whereas after 5 kyr BP the maximum uptake decreases and turns into outgassing after 3 kyr BP (keep in mind that PISCES has an equilibrium outgassing of around 0.5 GtC a$^{-1}$ that compensates the riverine carbon input).

*Acknowledgements.* J.S. would like to thank the dearly missed Ernst Maier-Reimer for his countless advice and help, Ernst's perpetual willingness to answer his questions and to discuss scientific issues even beyond office hours – at which time, however, discussions were preferably held not in the office but in more enjoyable surroundings, and strictly had to change subject after the third beer - with a bit of luck to his less known months-long journeys from Germany to India and the Saharan Desert by car. This work would not have been possible without him.

The authors acknowledge support by the German Research Foundation through the Collaborative Research Centre Climate-Biogeochemistry Interactions in the Tropical Ocean (SFB754) and the DFG project "Climate impact on marine plankton dynamics during interglacials" (Grant DFG SCH 762/3-1) and the Excellence Cluster Future Ocean (Grant FO EXC 80/1). We also wish to thank the NEMO/PISCES team for providing their models and general support. Computations were carried out on a NEC-SX-ACE at the computing centre of the Christian-Albrechts-University Kiel, Germany. Finally, the authors wish to acknowledge the use of the Ferret programme for analysis and graphics in this paper. Ferret is a product of NOAA's Pacific Marine Environmental Laboratory.

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

**Table 1.** Experiment names and characteristics. See also Fig. 1 for the temporal evolution of atmospheric greenhouse gases. Lower entries in column 'forcing-exp' indicate the KCM-experiments that have been used to force the BGC-experiments. (x10) indicates an acceleration factor of 10. $CH_4$ and $N_2O$ are not prescribed in PISCES.

| Experiment | Model | orbit [kyr BP] (forcing exp) | $pCO_2$ [ppm] | $pCH_4$ [ppb] | $pN_2O$ [ppb] | model years |
|---|---|---|---|---|---|---|
| KCM-CTL | KCM | 9.5 | 263.77 | 677.88 | 260.6 | 7,860 |
| KCM-HOLx10 | KCM | 9.5 - 0 (x10) | 263.77 -286.2 | 575 - 805 | 258 - 268 | 950 |
| KCM-HOL | KCM | 9.5 - 0 | 263.77 -286.2 | 575 - 805 | 258 - 268 | 9,500 |
| BGC-CTL | PISCES | KCM-CTL | 263.77 | n/a | n/a | 7,860 |
| BGC-HOLx10 | PISCES | KCM-HOLx10 | 263.77 -286.2 | n/a | n/a | 950 |
| BGC-HOL | PISCES | KCM-HOL | 263.77 -286.2 | n/a | n/a | 9,500 |

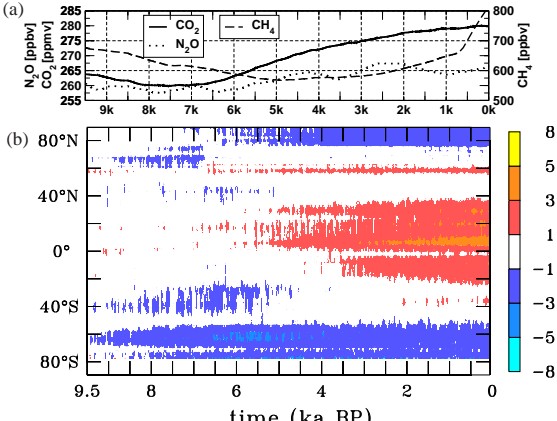

**Fig. 1.** Forcing for KCM-HOL and BGC-HOL experiments: **(a)** Holocene atmospheric greenhouse gas concentrations ($CO_2$, ppm; $CH_4$ and $N_2O$, ppb) derived from EPICA ice cores (Augustin et al., 2004) and provided by PMIP and **(b)** short wave radiative forcing at the sea/sea-ice surface in W m$^{-2}$ for the BGC-HOL experiment as computed in experiment KCM-HOL (i.e., the astronomical TOA changes over the Holocene as shown in Jin et al. (2014) filterd by ECHAM5, the atmospheric component of KCM). Hovmöller diagramme of the anomaly of zonal and annual mean since 9.5 kyr BP as 50 year running mean. Anomalies are derived by subtracting the average over the first 20 years from the annual mean values.

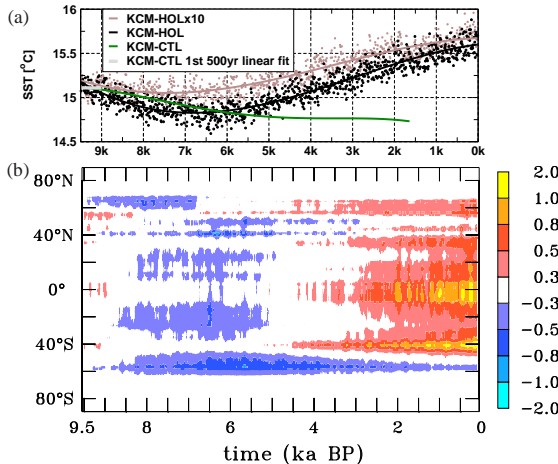

**Fig. 2. (a)** time series of annual and global mean SST in °C for the three KCM experiments KCM-HOL (non-accelerated forcing, black), KCM-HOLx10 (10 times accelerated forcing, brown), and the control experiment KCM-CTL (green). Circles represent annual averages (not every year shown), solid lines a 4th-order polynomial fit. The bold grey line indicates a linear fit over the first 500 years of experiment KCM-CTL. **(b)** Hovmöller diagramme of the zonal mean SST anomaly in °C for KCM-HOL, computed by subtracting the average over the first 20 years of KCM-HOL from annual mean values and smoothed by a 10 yr running mean. See colour bar for contour intervals.

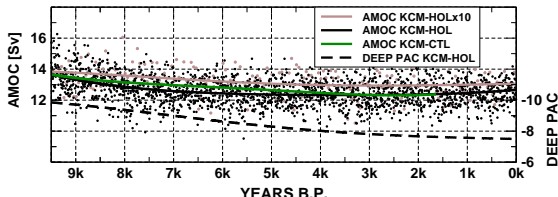

**Fig. 3.** As in Fig. 2(a), but for the maximum meridional overturning circulation in the Atlantic at 30°N (solid lines, left axis) and for the deep Pacific between 3000 m and 5000 m depth at 0°N (dashed line, right axis) in Sv ($10^6$ m$^3$ s$^{-1}$).

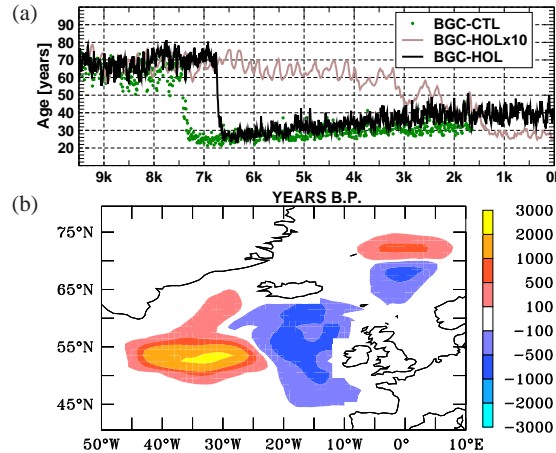

**Fig. 4. (a)** Idealised age (time since contact with the surface) in years averaged over a volume in the deep North Atlantic (40 °W - 10 °E, 40°N - 60°N, 1800 m - 2500 m depth) based on annual mean values for KCM-HOL (black), KCM-HOLx10 (brown) and KCM-CTL (green). **(b)** Change in annual maximum mixed layer depth in m in the North Atlantic between two periods before and after the shift in watermass age in KCM-HOL (7.8 minus 5.8 kyr BP, mean over 200 years centred around the respective dates).

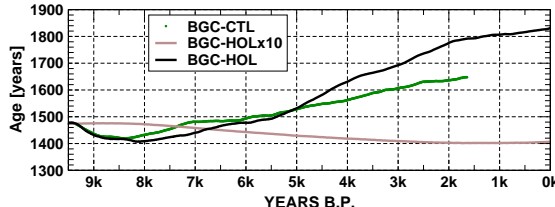

**Fig. 5.** As Fig. 4(a), but for the idealised age in years averaged over a volume in the deep North Pacific (150°E - 130°W, 40°N - 60°N, 2500 m - 3500 m depth).

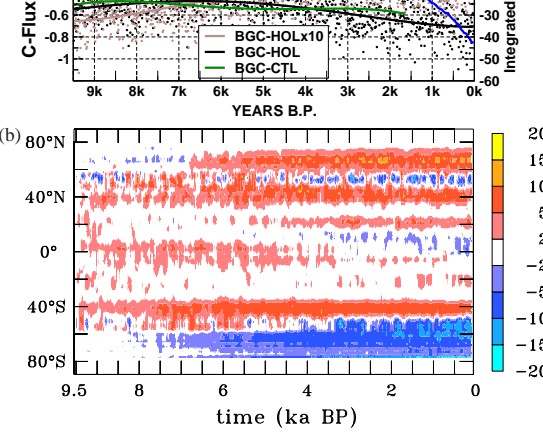

**Fig. 6.** As Fig. 2, but **(a)** for the global atmosphere-ocean carbon flux (GtC yr$^{-1}$) of experiments BGC-HOL (black), BGC-HOLx10 (brown) and BGC-CTL (green) and the integrated atmosphere-ocean carbon flux for BGC-HOL (GtC, blue). Negative values indicate oceanic outgassing. The net outgassing of around 0.5 GtC yr$^{-1}$ is balancing the river input of carbon. **(b)** Hovmöller diagramme of the zonal mean atmosphere-ocean carbon flux change (mol C m$^{-2}$ s$^{-1}$) of experiment BGC-HOL.

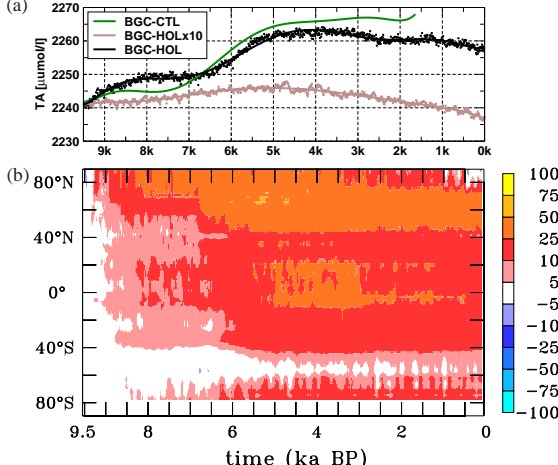

**Fig. 7.** As Fig. 2, but for **(a)** time series of global mean total alkalinity (TA) at the surface in $\mu$mol l$^{-1}$ as 8th-order polynomial fit, and **(b)** Hovmöller diagramme of the BGC-HOL changes in zonal and annual mean surface TA in $\mu$mol l$^{-1}$.

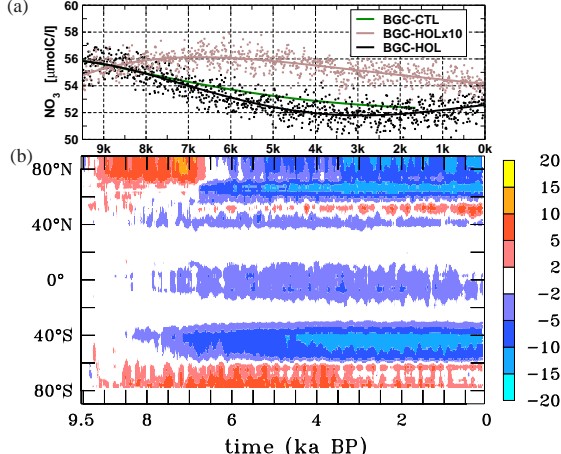

**Fig. 8.** As Fig. 2, but **(a)** for the average $NO_3$ concentration averaged over the uppermost 100 m in $\mu$mol C l$^{-1}$ and **(b)** Hovmöller diagramme of the BGC-HOL changes in zonal and annual mean $NO_3$ concentration in the upper 100 m in $\mu$mol C l$^{-1}$.

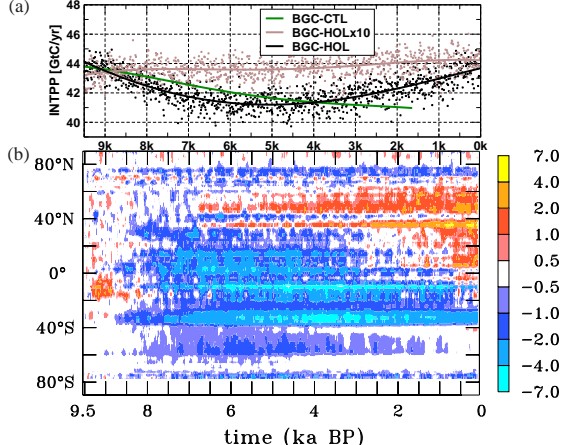

**Fig. 9.** As Fig. 2, but **(a)** for time series of global primary production integrated over the upper 100 m in GtC yr$^{-1}$ (INTPP), and **(b)** Hovmöller diagramme of the BGC-HOL changes in zonal and annual mean INTPP in mol C m$^{-2}$ s$^{-1}$ x $10^8$.

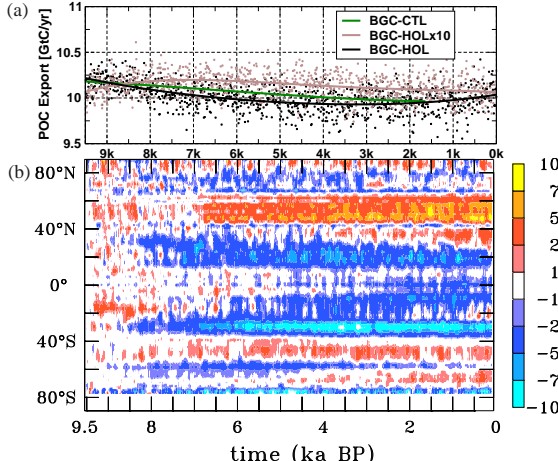

**Fig. 10.** As Fig. 2, but **(a)** for time series of global export production at the bottom of the euphotic layer in GtC yr$^{-1}$, and **(b)** Hovmöller diagramme of the BGC-HOL changes in zonal and annual mean global export production in mol C m$^{-2}$ s$^{-1}$ x 10$^9$.

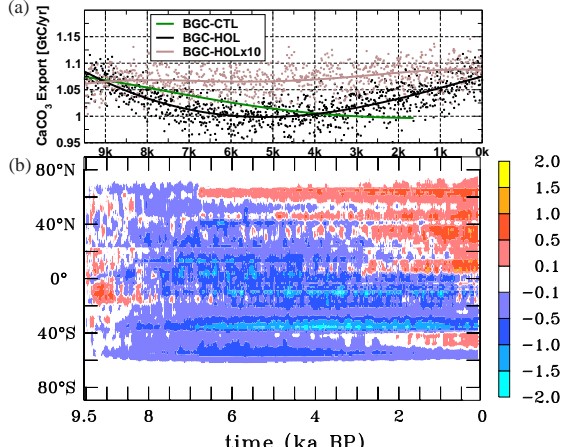

**Fig. 11.** As Fig. 2, but **(a)** for time series of global CaCO$_3$ export production at the bottom of the euphotic layer in GtC yr$^{-1}$, and **(b)** Hovmöller diagramme of BGC-HOL change in zonal and annual mean CaCO$_3$ export in mol C m$^{-2}$ s$^{-1}$ x 10$^9$.

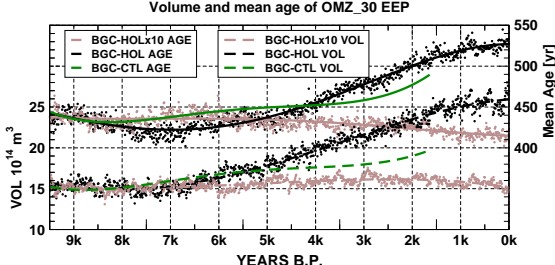

**Fig. 12.** Time series of Eastern Equatorial Pacific OMZ-volume for a threshold of 30 $\mu$mol l$^{-1}$ in 10$^{14}$ m$^3$ (left axis, lower dashed curves) and mean age of water mass in the EEP OMZ in years (right axis, upper solid curves). EEP defined as 140°W - 74°W, 10°S - 10°N, 0-1000 m depth. Circles represent annual mean values of the OMZ-volume, dots are annual mean values of the water mass age in the EEP OMZ for BGC-HOL (black), BGC-HOLx10 (brown) and BGC-CTL (green). Solid lines represent polynomial fits of 4th order.

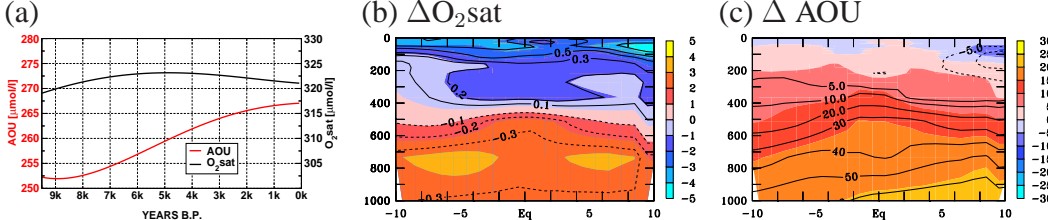

**Fig. 13. (a)** time series of AOU and $O_2$-sat in the EEP (region as in Fig. 12, but for 100 - 800 m depth), and **(b, c)** meridional sections of zonal mean differences of late Holocene minus early Holocene for **(b)** $O_2$-sat (shading, $\mu$mol l$^{-1}$) and temperature (contours, °C) and **(c)** AOU (shading, $\mu$mol l$^{-1}$) and water mass age (contours, years). Contour interval for temperature is 0.1 from -0.3 to 0.3°C, with additional lines for 0.5, 0.7, and 1°C. Contour interval for water mass age is 10 yr from -10 to 60 yr, with additional lines at -5 and 5 yr. Zero contours are omitted.

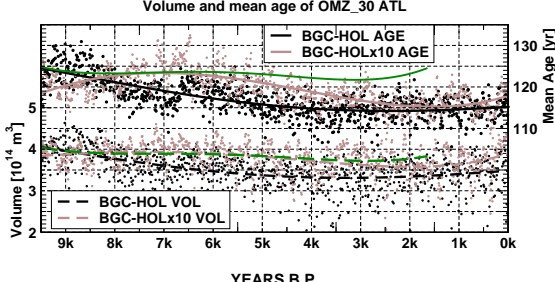

**Fig. 14.** As Fig. 12 but for the Atlantic in the area of 5°W - 15°E, 30°S - 5°N, 0 - 1000 m depth. OMZ-volume for a threshold of 30 $\mu$mol l$^{-1}$ in 10$^{14}$ m$^3$ and water mass age in years.

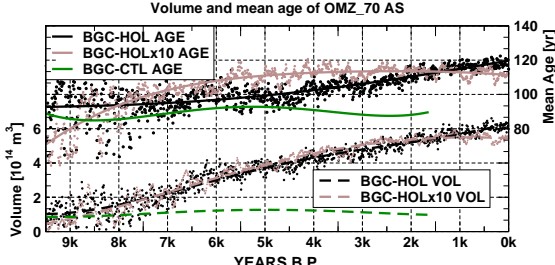

**Fig. 15.** As Fig. 12 but for the Arabian Sea in the area of 55°E - 75°E, 8.5°N - 22°N, 100 - 800 m depth. OMZ-volume for a threshold of 70 $\mu$mol l$^{-1}$ in 10$^{14}$ m$^3$ and water mass age in years.

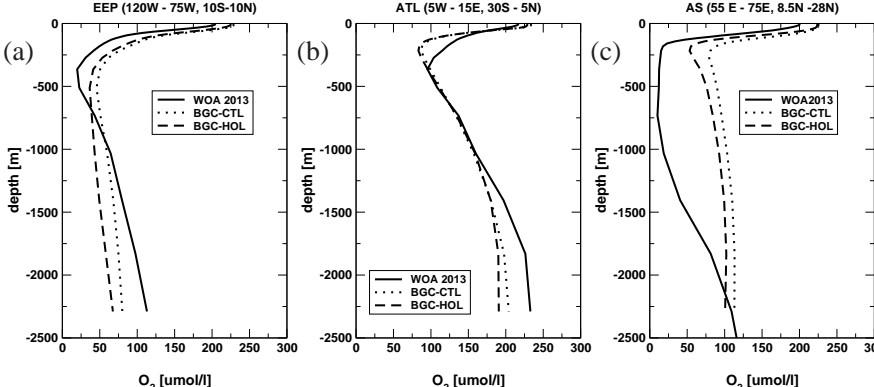

**Fig. A.1.** Simulated and observation-based profiles of average $O_2$-concentration in $\mu$mol l$^{-1}$ in the three major oxygen minimum zones in the world ocean for **(a)** the eastern equatorial Pacific, **(b)** the tropical South Atlantic, and **(c)** the Arabian Sea. Based on observations (WOA2013, Garcia et al., 2013, solid), and from experiments BGC-CTL (dotted) and BGC-HOL (dashed) averaged over the last 200 years.

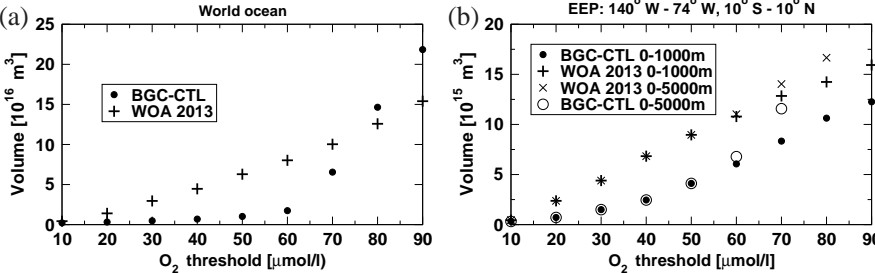

**Fig. A.2.** Simulated and observation-based (WOA2013, Garcia et al., 2013) volume of water masses with oxygen concentration below the threshold value indicated on the x-axis for **(a)** the world ocean from the surface to the seafloor in $10^{15}$ m$^3$, and **(b)** the eastern equatorial Pacific for 0 - 1000 m and 0 - 5000 m in $10^{14}$ m$^3$. See legends for explanations of symbols.

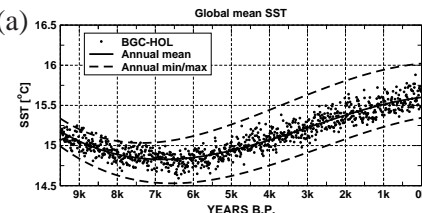
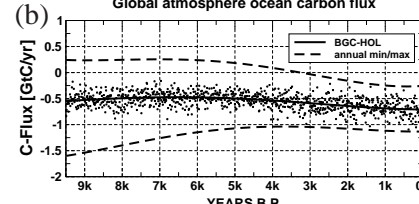

**Fig. A.3.** Time series of **(a)** SST and **(b)** atmosphere-ocean carbon flux (negative upward) for experiment BGC-HOL (black dots: annual mean values, solid black line: 4th-order polynomial fit), and annual minimum and maximum (dashed black lines, 4th-order polynomial fits), indicating the range of the annual cycle.

