# Peer review of "Climate and marine biogeochemistry during the Holocene from transient model simulations"

_Biogeosciences, 2017_

## Referee Comment (RC1) · Anonymous Referee #1 · 9 Feb 2018

This manuscript describes the results of a transient simulations with a comprehensive atmosphere-ocean-sea ice biogeochemistry model over the past 9.5 kyr of the current interglacial. A major finding is a large change in the oxygen minimum zones over the Holocene. The authors also discuss how marine changes may alter atmospheric $CO_2$ concentration. Such long transient simulations are rare and as such this manuscript should be published after the following comments have been addressed.

1) The role of model drift should be discussed a bit more in depth. A ctrl run of 2 kyr seems somewhat short and there may be still some long-term adjustments going on. Can the authors exclude that model drift leads to the changes in Pacific OMZ,

overturning, water mass age and related biogeochemical changes? If I understand it correctly, SST in the CTRL is expected to change by 0.3 K from 9.5 to 6.5 kyr BP which is substantial compared to the signal in the transient run.

2) The biogeochemistry model is run offline. How is convection and diffusion and related tracer transport treated? Are water mass age in online and offline model versions the same?

3) The vertical resolution is 31 layer in the ocean. How many layers are below the mixed layer and what is the range in layer thickness. I suspect that layer thickness in the deep may be very large which may be problematic for deep Pacific oxygen?

4) While the representation of physical processes and of atmospheric dynamic is advanced in the online model compared to the current crop of Earth System Models of Intermediate Complexity (typically used to explore the carbon-climate system over the Holocene), major biogeochemical processes are neglected which likely affects results. These processes should be explicitly mentioned and their potential influence on results discussed. It is know that shallow water carbonate deposition changed over the Holocene (Vecsei and Berger, 2004) . Ocean-sediment interactions and changes in weathering and their legacy effects are missing. How is the freshwater input into the North Atlantic for the 8.2 kyr event represented? What is the role of changes in land biosphere carbon inventory? Is there an influence from volcanic eruptions (Huybers and Langmuir, 2009) In this respect, insight from earlier ocean modelling studies on Holocene $CO_2$ evolution are relevant (e.g.; (Brovkin et al., 2016;Menviel and Joos, 2012;Ridgwell et al., 2003;Broecker et al., 2001))

5) Unfortunately, atmospheric $CO_2$ was prescribed in the biogeochemistry module of the model. This makes the interpretation of air-sea carbon fluxes very difficult and it is difficult if not impossible to quantify how the atmospheric $CO_2$ would evolve in the coupled model system with free atmospheric $CO_2$. I find Figure 7 to be highly misleading. It is fine to show the time-integrated air-sea flux, but to equate this with

atm. CO2 changes is not possible.

In case the ocean would be in steady-state in terms of circulation, prescribing atmospheric CO2 to the ocean would lead to an outgassing when atmospheric CO2 is decreasing (9.5 to 7 kyr BP) and to an ingassing of carbon into the ocean when atm. CO2 is increasing. The simulated air-sea carbon flux under varying climate must be interpreted in the context of this atm. concentration driven flux. There is no change in the ocean's carbon inventory in the first few millennia despite a decrease in the prescribed CO2. The model ocean would lose carbon to the atmosphere if atm. CO2 would be freely evolving. The model ocean takes up carbon during the time of the atm. CO2 rise (7 to 4.5 kyr). Is this uptake driven by the prescribed concentration or is a concentration driven uptake partly mitigated by the modelled changes? I have the impression that the uptake is rather small for the magnitude of the CO2 increase. In the late Holocene, atm. CO2 varies little and the ocean is loosing CO2. So I guess overall the climate driven ocean signal is likely an outgassing of CO2 to the atmosphere over the entire Holocene. Unfortunately, this is hard to verify with the current setup.

Figure 7 should be deleted and the cumulative air-sea flux should be shown in Figure 6. The interpretation of the cumulative air-sea flux must be revised.

6) It would be appreciated if changes in O2 and other tracers would be a bit better attributed to underlying processes. There is hardly any quantitative attribution of O2 changes to mechanisms. How large is the influence of solubility changes versus changes in remineralization/water mass age? It would be good to separate solubility-driven changes and changes in –AOU, i.e. biologically-mediated changes. Do the authors also find the conventional anti-correlation between solubility-driven and oxygen-utilization-driven changes in the OMZ (Bopp et al., 2017)?

How does stratification change and what is its role for O2 and BGC evolution?

7) P.24, l2: export production is the wrong metric to judge the efficiency of the biological pump. What matters for atmospheric CO2 is nutrient utilization. The interpretation

given here is flawed.

Further comments 1) P4, 2. Para: Other transient simulations include those from the Trace21kyr experiment by NCAR and as mentioned above there are quite a few EMIC studies available that are worth mentioning.

2) P9: Are changes in ice albedo be taken into account? Please mention here that solar TSI and volcanic forcing is neglected

3) P9 The Indermühle $CO_2$ data have been update and the chronology adjusted. See e.g. Elsig et al.. Which specific $CO_2$ data and age-scale is used to force the model?

4) P12: 1 para would better fit in method section

5) P12, l21: delete "again"

6) P12, l25; the drift appears not "modest"; please delete modest.

7) P13, line 1-2: It seems that the whole signal may be explained by drift?

8) P13, sec 3.1.2 It would be useful also to discuss Indo-Pacific overturning here.

9) P14: line 18 to 21: How does this increase in water mass age from 1500 to 1800 yr relate to model drift? What is driving this change in Indo-Pacific water mass age and overturing circulation? How do the age change compare in the online and offline model version?

10) P18, line 28: What is the role of saturation/solubility changes?

11) P20, discussion: The authors should say something here and in the results about the SST evolution at different seasons. See for example discussion on Holocene temperature evolution by (Liu et al., 2014;Samartin et al., 2017).

12) How is the control extrapolated in Figure 5?

References

Bopp, L., Resplandy, L., Untersee, A., Le Mezo, P., and Kageyama, M.: Ocean (de)oxygenation from the Last Glacial Maximum to the twenty-first century: insights from Earth System models, Philosophical Transactions of the Royal Society A: Mathematical, Physical and Engineering Sciences, 375, 2017. Broecker, W. S., Lynch-Stieglitz, J., Clark, E., Hajdas, I., and Bonani, G.: What caused the atmosphere's $CO_2$ content to rise during the last 8000 years?, Geochemistry, Geophysics, Geosystems, 2, 2001GB000177, 2001.

Brovkin, V., Brücher, T., Kleinen, T., Zaehle, S., Joos, F., Roth, R., Spahni, R., Schmitt, J., Fischer, H., Leuenberger, M., Stone, E. J., Ridgwell, A., Chappellaz, J., Kehrwald, N., Barbante, C., Blunier, T., and Dahl Jensen, D.: Comparative carbon cycle dynamics of the present and last interglacial, Quat. Sci. Rev., 137, 15-32, https://doi.org/10.1016/j.quascirev.2016.01.028, 2016.

Huybers, P., and Langmuir, C.: Feedback between deglaciation, volcanism, and atmospheric $CO_2$, Earth and Planetary Science Letters, 286, 479-491, DOI: 10.1016/j.epsl.2009.07.014, 2009.

Liu, Z., Zhu, J., Rosenthal, Y., Zhang, X., Otto-Bliesner, B. L., Timmermann, A., Smith, R. S., Lohmann, G., Zheng, W., and Elison Timm, O.: The Holocene temperature conundrum, Proceedings of the National Academy of Sciences, 2014.

Menviel, L., and Joos, F.: Toward explaining the Holocene carbon dioxide and carbon isotope records: Results from transient ocean carbon cycle-climate simulations, Paleoceanography, 27, PA1207, 10.1029/2011pa002224, 2012.

Ridgwell, A. J., Watson, A. J., Maslin, M. A., and Kaplan, J. O.: Implications of coral reef buildup for the controls on atmospheric $CO_2$ since the last glacial maximum, Paleoceanography, 18, 10.1029/2003PA000893, 2003.

Samartin, S., Heiri, O., Joos, F., Renssen, H., Franke, J., Bronnimann, S., and Tinner, W.: Warm Mediterranean mid-Holocene summers inferred from fossil

midge assemblages, Nature Geosci, advance online publication, 10.1038/ngeo2891 http://www.nature.com/ngeo/journal/vaop/ncurrent/abs/ngeo2891.html#supplementary-information, 2017.

Vecsei, A., and Berger, W. H.: Increase of atmospheric $CO_2$ during deglaciation: constraints on the coral reef hypothesis from patterns of deposition, Global Biogeochem. Cycles, 18, 1-7, 2004.

---

## Referee Comment (RC2) · Anonymous Referee #2 · 11 Feb 2018

This paper provides a very worthy contribution to this special issue in honour of Ernst Maier-Reimer, who himself was a pioneer in modelling of global ocean biogeochemical cycles.

In this paper the Kiel Climate Model, coupled to PISCES is forced by accelerated and non-accelerated orbital parameters and atmospheric CO2 for the last 9,500 years. This paper is the first to report on changes in the strength of the carbon pumps that drive the ocean-atmospheric CO2 flux and dynamics of oxygen in seawater, including the oxygen minimum zones, in response to these forcings. The authors state as the most significant result that they find a substantial increase in the volume of the eastern equa-

torial oxygen minimum zone into the late Holocene, but only in the non-accelerated simulation, concluding that non-accelerated experiments are required for analyses of marine biogeochemistry in the Holocene. One obvious question would be whether this conclusions extrapolates further back in time (glacial-interglacial timescales; ocean anoxic events...).

The manuscript is well written, and an important contribution to improve our knowledge of the processes involved with carbon and oxygen cycling in the ocean. Below are some comments /suggestions, followed by minor typos: 1. In the conclusion the authors acknowledge the fact that most ocean-atmosphere coupled models do not simulate the mid-Holocene climate optimum under the applied astronomical and CO2 forcings, and suggest that perhaps full scale ESM, including a land biosphere and free carbon cycle, may resolve this. Although the authors have been very careful in their wording, I do wonder whether if would be helpful to describe their model simulations as sensitivity tests to certain forcings over this time period. This doesn't take away the novelty of the results, but emphasizes the limitations. 2. In the experiment set-up only CO2 is allowed to change, whereas methane and nitrous oxide concentrations were kept constant. According to Fluckiger et al. (2002) especially methane fluctuated considerably more during the Holocene than CO2. Would this not influence the greenhouse forcing? 3. Are planetary and cloud albedo included in the radiation calculations? 4. Comparison with proxy reconstructions is a bit thin: Inferences of AMOC: it would be nice to see the model simulations compared with proxy reconstructions (for example: Hillaire-Marcel et al., 2001; Hoogakker et al., 2011, 2015; Thornalley et al., 2013). Volumes of oxygen minimum zones: while the authors refer to the review of Moffit et al. (2015), it would have been nice to see how changes in oxygen concentrations compare with local continental margin Holocene nitrogen isotope records of for example the Arabian Sea and eastern Pacific Ocean.

Minor comments: P 15: line six, should this be -0.4 GtC?yr? P 16: line 20, double relevance. P 19: line 24: sea-ice not seaice. P 22: line 28: affect instead of effect,

or impact? P 24: line 23: physical instead of physical, line 24: extremes rather than extrema?

---

## Author Comment (AC1) · 7 Mar 2018

**Response to Rev#1**

We wish to thank Rev#1 for his many thoughtful comments and in particular for pointing us to the work of Liu et al. (2014) on the Holocene climate conundrum. We performed additional analyses to address the reviewer's comments where possible and also extended the control experiments KCM-CTL and BGC-CTL further. Comments are adressed point-by-point in the following. In addition we plan to divide the Discussion into subsections for better readability as it was already fairly lengthy and will now have

to further increase in length.

**1) Role of model drift**

In deed the control run of 'only' 2000 years turned out to be non-satisfactory for the non-accelerated experiments, and Rev#1 is right in that we could not exclude that at least part of the signals we discuss are remaining model drift or internal model variability. 2000 years seemed quite long with regard to the accelerated 950 year-long experiments. It was then unexpected that there was still some model drift (and/or internal variability) that amounted, for some parameters, to variability of similar magnitude as the variations of the non-accelerated transient experiments.

The basis for the KCM experiments is a 1,000 year KCM experiment with 0k orbital parameters, 286.6 ppm $CO_2$, and 805 ppbv $CH_4$ concentration (with a final global average SST of 15.8C), followed by a spin up for a further 1,000 years with 9.5kyr BP orbital and GHG forcing. From this state the KCM-CTL and KCM-HOL experiments were started. Apparently the 2x 1,000 year spinup time was still not long enough, and a model drift remained in KCM-CTL that was significant given the small temperature changes that occured during the Holocene. We have become aware of this problem during the analyses of experiment KCM-HOL, and extended the control runs (KCM-CTL and BGC-CTL) since, but they are not finished yet and will still need up to 6 more month to do so (currently at 6200 years, 3.2 kyr BP).

From the current state, this has an impact of the interpretation of the SST (revised Fig. 2a, see also response to FC 7), but not on the finding about the EEP OMZ (revised Fig. 13). For the deep North Pacific, also in the control run the ideal age increases, but not as strong as in BGC-HOL (100 years younger at 3.2 kyr BP). For the Eastern Equatorial Pacific OMZ, however, the extended control run does not show the increase

in volume as simulated by BGC-HOL, so that we are now more sure that the results are not caused by model drift.

We also note that for all the paleo experiments in the literature, there are no(!) control experiments to be found for transient and time-slice experiments (Fischer and Jungclaus, 2011; Varma et al., 2016; Liu et al., 2014; Bopp et al., 2017). So this is a more general problem in the paleo climate modelling community, probably based on an underestimation of model drift and internal model variability compared to the relatively weak forced Holocene variability on long time scales.

We will include the extended control run in our revised manuscript and revise the text were necessary.

**2) Off-line biogeochemistry**

Convection and diffusion are simulated as in the online version, based on the stored mixing parameters from OPA9 as part of KCM. So there might be some differences from the monthly time-averaging of the online output and the non-linearity of mixing, but we have not investigated this systematically for our model configuration as there is no online version of KCM-PISCES available.

For the same reason, we can not make a statement about potential differences of water mass ages in the on- and off-line versions of PISCES.

**3) Vertical resolution of the ocean**

There are 20 layers below 100m depth, and layer thickness near the bottom is about 500m. We have no indication that this causes problems for the simulation of deep

Pacific $O_2$ within the general limitations due to the coarse model resolution.

We will describe the vertical resolution of the ocean model in the model description.

**4) Missing processes, potential impact on results should be discussed**

- sediment is not included in model setup

- no changes in weathering

- no coral reefs (Vecsei and Berger, 2004)

In setting up our experiments, we followed the standard PMIP protocol. So while the above processes are not included in the model setup, their combined effects on GHG climate forcing must be represented by the reconstructed PMIP GHG forcing from ice cores. With regard to the evolution of atmospheric $CO_2$, we plan to point to the study of Brovkin et al. (2016), were all these processes are described in fair detail and nicely summarized in their Fig. 8.

- how is freshwater pulse of 8.2 kyr BP reprented in the forcing?

The freshwater pulse at 8.2 kyr BP is not included in the forcing, and hence we can not expect to find changes in AMOC related to it. This will be picked up in the discussion of AMOC (Sec. 3.1.2 and Discussion).

- role of changes in the land biopshere carbon inventory

The land biosphere is not included in the model, but we use reconstructed $pCO_2$, which should include any contribution of the land biosphere to atmospheric $pCO_2$. So what remains neglected is albedo changes from changes in vegetation, but we can not quantify the potential impact this would have on the simulated climate.

- potential influence from volcanic eruptions?

Including volcanic aerosol forcing would likely have a cooling effect during the first 1-2 years following the volcanic eruptions, but this is difficult to quantify for KCM without performing the experiment. For the same reason, also any integrated effect of volcanic eruptions on long term evolution of temperature is difficult to establish, but would likely to be small based on the coupled climate model experiments for the last millennium (Brovkin et al., 2010) that indicate a -0.8cooling for the 1258 eruption (the largest eruption during the simulated period), but within a decade surface air temperature fluctuations are within the background range (their Fig. 1). Any effect on atmospheric $pCO_2$ related to post-glacial increased volcanic activity and additional outgassing of 1,000-5,000 $GtCO_2$ between 12 kyr and 7 kyr BP (Huybers and Langmuir, 2009) will be included in the prescribed PMIP GHG-forcing (which shows decreasing $pCO_2$ during the early Holocene).

In summary, while these are all interesting points, we do not want to hypothesize about the potential effects of the omitted forcings and model components.

Also in response to Rev#2, we will now describe our model experiments as sensitivity experiments to the PMIP orbital and GHG forcings with likely deviations from the

Holocene climate variations that are caused by other forcings and biogeochemical processes that are not included in our model setup. We will explicitly mention the neglected components and forcings, and indicate where this might have a direct impact on our results.

**5) Interpretation of $CO_2$ fluxes in the light of prescribed $pCO_2$**

We think that we were quite careful in our wording as to how much the diagnosed $CO_2$-fluxes can be meaningful for a quantification of a potential contribution of the ocean to atmospheric $pCO_2$, however, seemingly not careful enough.

We will delete Fig. 7 and include the time-integrated ocean-atmosphere carbon flux in a revised Fig. 6a as suggested. We will revise the description of the time-integrated carbon flux along the following lines:

Early Holocene: The integrated carbon flux is constant (revised Fig. 6), despite a decrease in atmospheric $CO_2$ (revised Fig.1a): During this initial phase, the global mean SST in KCM-HOL is decreasing by about 0.3C (orig. Fig. 2a), and the alkalinity in BGC-HOL is increasing by 10 $\mu$mol/l (orig. Fig. 8a), while export production is slightly decreasing by 0.2 GtC/a (orig. Fig. 11a). In summary, these effects balance out during the early Holocene, and the time-integrated ocean-atmosphere carbon flux remains about zero until 7 kyr BP (revised Fig. 6a, blue curve).

In the mid-Holocene, from 7 kyr BP to 4 kyr BP, the time-integrated carbon flux is slowly increasing to a total ocean uptake of 10 GtC. This occurs during a period of atmospheric $pCO_2$ increase that would drive such a flux, and also SST is rising slowly, while EP is further decreasing and alkalinity is increasing by a further 10 $\mu$mol/l.

In the late Holocene, after 4 kyr BP, the ocean is outgassing a total of 50 GtC, potentially driven by the simulated increase in SST and damped by the increasing prescribed atmospheric $pCO_2$.

In summary, the contribution of oceanic processes to air-sea carbon fluxes is in line with the prescribed $pCO_2$ in the mid-Holocene, while reversing the flux expected from the atmospheric forcing in the early and late Holocene.

**6) Attribution of changes of $O_2$ and other tracers to underlying processes**

- stratification changes and impact on $O_2$

In contrast to studies based on global warming and $O_2$ (Cocco et al., 2013; Bopp et al., 2013) and marine biological production (Steinacher et al., 2010), that showed an impact of stratification changes on $O_2$ fields and marine biological production, here we simulate much smaller temperature variations. Global and annual mean of the MLD in KCM-HOL reveal little temporal variability: MLD is around 48m at 9.5kyr BP, and starts to decrease after 5.5 kyr BP to around 47m at 0 kyr BP. We, therefore, state that stratification changes play only a minor role for $O_2$ changes during the Holocene. We will include this in the Discussion.

-$O_2$ saturation and AOU

We computed also the fields of $O_2$-saturation and AOU for experiment BGC-HOL. While $O_2$-saturation has a similar evolution as SST, AOU reflects the slow-down in circulation in the Pacific (see Fig??,left). Meridional sections reveal a different Holocene-trend of $O_2$-saturation in the upper and lower part of the OMZ: decrease in the upper part,

increase below 400m, reflecting the evolution of temperature. AOU is stronger in the late Holocene at all depths save from a small decrease in the upper 100m off the equator. The long term changes show a decoupling of O2sat and AOU, the former driven by temperature changes, the latter by a slower circulation (increase in water mass age).

This differs from the shorter time-scale fluctuations that show compensating effects as in Bopp et al. (2017), where a decrease in $O_2$-saturation has a signature in lower AOU and vice versa.

We will mention the above explicitly in the revised ms and will add a new figure similar to the above. We further refer to the already existing attribution of the changes in the EEP OMZ to changes in circulation as represented by the ideal age tracer, and the fairly constant export production in the EEP OMZ region (Section 3.2.5).

**7) export production wrong metric to judge efficiency of the biological pump**

With due respect, on this point we disagree with Rev#1. A large number of studies describe the biological carbon pump, also called soft tissue pump as driven by the export production (e.g. Six and Maier-Reimer, 1996; Sigman and Hain, 2012; Ducklow et al., 2001), and the whole JGOFS project was based on this principle. See also https://www.us-ocb.org/biological-pump/ for a very condensed description. Evidently the export production is related to the uptake of nutrients in the euphotic zone, but as not all of the nutrients that are taken up in the euphotic zone are exported to depth because of remineralization and grazing in the euphotic layer itself, we emphasize that export production is a correct metric for the strength of the biological carbon pump.

We do not intend to change the section in question, but will include a reference to the biological pump.

**Response to further Comments (FC)**

**FC 1) Other transient simulations not mentioned (p4, 2. para)**

We referenced some exemplary transient simulations rather than trying to give a complete overview (again, our focus being more on the marine biogeochemistry), but to address this point we will include references to Brovkin et al. (2016) as an example of EMIC experiments, and to Liu et al. (2014) as an example for the 21ka experiment, and to Fischer and Jungclaus (2011) as a further non-accelerated coupled model experiment from mid-to-late Holocene (looking at changes in the seasonal SST cycle).

**FC 2) Are changes in ice albedo taken into account? (p9)**

Changes in sea ice cover are simulated by LIM, the sea ice component of NEMO, but that is probably not meant here. We will mention explicitly in Section 2.2 that we do not take into account solar TSI and volcanic forcing, nor changes in the continental ice sheets (neither topography nor albedo) and also no fresh water pulses.

**FC 3) Forcing data GHG**

We admit somewhat shamefacedly that due to a misunderstanding we stated that we force KCM-HOL with only $CO_2$ as green house gas. However, the experiments were

also forced with transient $CH_4$ and $N_2O$ from the PMIP data base. This mistake in the description of the forcing, however, does not change the findings of our study.

We will rewrite section 2.2.1 accordingly and revise Fig. 1a to include the time series of $CH_4$ and $N_2O$ (see above). We will also provide the internet address from where we obtained the data, and add the reference to Augustin et al. (2004) decribing the EPICA data.

https://www.paleo.bristol.ac.uk/ ggdjl/pmip/pmip_hol_lig_gases.txt

**FC 4) 1st para would better fit in methods sections (p12, ln 2)**

Since this para describes the common features of the following figures, we feel that this para is well-placed at the beginning of the Results section, in particular having in mind that readers might skip the Methods and go straight to the Results section.

However, it could be moved to a new section 2.5 Common features of figures in results section in the Methods section but we would prefer to leave it at its current position.

**FC 5) delete 'again' p12, ln 21**

will be deleted

**FC 6) drift is not 'modest', please delete modest (p12, ln 25)**
we will delete 'modest'

**FC 7) It seems the whole [SST] signal may be explained by drift? (p13, ln1-2)**

See also response to main comment **1)**. As we have extended experiment KCM-CTL in the meantime by a further 4800 years (leading up to 3.2 kyr BP) we are in a position to state that parts of the initial SST decrease in KCM-HOL can indeed be explained by the drift (the decrease is stronger in KCM-HOL), while the following SST increase is damped by the model drift (which becomes smaller after 6 kyr BP). As a result, the initial SST decrease would be weaker in a drift-free setup, while the following SST increase would be stronger. It would of course be ideal to run the extended control experiment until 0k, but that would potentially delay publication by several months (a minimum of 2.5 months, from past experience more likely 4-6 months).

We will revise Section 3.1.1 accordingly, and ammend Fig. 2a to include the extended control run as far as possible.

**FC 8) Indo Pacific overturning should be discussed also (p13, sec. 3.1.2)**

In response to FC8 we also analysed the Indo Pacifc overturning. We found that the deep inflow into the Pacific decreases, e.g. at 3000m depth, 10S zonal mean MSF from 5 to 2 Sv around 7.5 kyr BP and remains about constant thereafter (based on 4th order polynomial fit to monthly data). In the upper ocean there are only minor variations. Also in the Indian Ocean the overturning is rather constant.

[Figure]

We will mention this in the revised ms., but do not intend to show an additional figure.

**FC 11) Authors should say sth. on SST evolution at different seasons (p20, Discussion)**

While we feel that this goes a bit beyond the scope of our manuscript (and possibly the focus of Biogeosciences), we analysed the summer (JJA) and winter (DJF) SST separately. Only when looking at the northern hemisphere poleward of 30N, we find some separation of the evolution of the annual mean and the JJA SST, with a later and longer lasting minimum (around 6 kyr BP and a major increase only after 3 kyr BP), with similar SST in early and late Holocene, and an amplitude of around 0.5 K. For the global mean SST, the JJA mean evolves similar as the annual mean SST. Also using the yearly maximum temperature, indicating local summer, does not change the temporal behaviour, it only shifts the SST curve upward by roughly 2 K.

With regard to the work of Samartin et al., 2017, that focusses on the Mediteranean, we find only very locally in the Adriatic (close to the data point 3 in Samartin et al) and in the orbitally only forced experiment a warmer mid-Holocene than late Holocene SST.

SST and land surface temperature might develop differently, and this might be investigated further using a 'site-stacked' approach as in Liu et al. (2014) in a future study aiming at KCM-proxy comparisons. As a first step, we computed the global and annual mean land surface temperature, and found a principally similar temporal evolution as for SST. We will work on the topic, also in the follow up of the study of Schneider et al. (2010).

**BGD**

With regard to the focus of our present study, however, and the limited relevance of the additional analysis for the investigated biogeochemistry, we would prefer not to include details of the above in the revised ms. but merely plan to add that the simulated SST evolution in KCM-HOL is seemingly not very sensitive to the choice of season. We further plan to point to the study of Liu et al., 2014 for a more detailed analysis of the general proxy-model mismatch for MH temperatures and its seasonal dependency.

**FC 12) How was BGC-CTL extendend (Fig. 5)**

BGC-CTL was extended by forcing PISCES for another cycle of the 2000 yrs available from KCM-CTL. Admittedly not an ideal approach, and this will be replaced by the extended BGC-CTL, based on the extended KCM-CTL in a revised Figure 5.

Further errors found by the authors

OMZ-volume was erroneously divided by 100, now correct volumes are given

Typo in Ref of (Leduc et al., 2010): Ma/Ca was corrected to Mg/Ca

E.g. misplaced p23 ln 3

**References**

[revised manuscript text omitted]

GB1035, doi:10.1029/2003GB002147, 2004.

[Figure]

**Volume and mean age of OMZ_30 EEP**

Legend:
- BGC-HOLx10 AGE
- BGC-HOL AGE
- BGC-CTL
- BGC-HOLx10 VOL
- BGC-HOL VOL
- BGC-CTL VOL

Y-axis (left): VOL $10^{16}$ m$^3$
Y-axis (right): Mean Age [yr]
X-axis: YEARS B.P.

**Fig. 1.** Revised Fig. 13

[Figure]

**Fig. 2.** Revised Fig. 1a

[Figure]

**Fig. 3.** Revised Fig. 6a

(a)    (b)    (c)

[Figure]

**Fig. 4.** New Fig. (a) AOU and O2sat in EEP. b) O2sat Late Holocene minus Early Holocene
zonal means EEP, c) as b) but for AOU

[Figure]

**Fig. 5.** Revised Fig. 2a

[Figure]

**Fig. 6.** Revised Fig. 5

---

## Author Comment (AC2) · 7 Mar 2018

**Response to Rev#2**

We wish to thank Reviewer#2 for the kind words and thoughtful comments that we intend to address as outlined below. In addition we plan to divide the Discussion into subsections for better readability as it was already fairly lengthy and will now have to further increase in length.

**1) Mismatch with observations/regard experiments as sensitivity study**

We find that a particularly useful suggestion of Rev#2 that we are happy to follow. Also, Rev#1 pointed us to the study of (Liu et al., 2014, The Holocene temperature conundrum), who investigated this mismatch of proxy-based warmer and model simulated colder mid-Holocene than late-Holocene in some detail. They found this to be a consistant feature for the three investigated coupled climate models. So this is a more widespread issue that we can not resolve here. See also response to Rev#1 FC11.

We will revise the Abstract, Introduction, and Discussion to describe our study as a sensitivity experiment to the prescribed forcing, with known deviations to the Holocene climate evolution estimates from proxy data.

**2) Relatively large $\text{CH}_4$ variations during the Holocene but $\text{CH}_4$ not included in forcing**

We admit that due to a misunderstanding between authors we stated that KCM was forced with only  $CO_2$  as time-varying greenhouse gas. However, the experiments were in fact also forced with transient  $CH_4$  and  $N_2O$  according to the PMIP protocol (https://www.paleo.bristol.ac.uk/ ggdjl/pmip/pmip\_hol\_lig\_gases.txt).

We apologize for this error, and will rewrite section 2.2.1 accordingly. We will also add the data source, and the reference to Augustin et al. (2004) that describes the EPICA ice core as data source for the Holocene greenhouse gas concentrations.

BGD
Since the CH4 variations were relatively large (about 100ppbv leaving aside the land use change related increase during the last 500 years), however, it still seems useful to estimate the effect of Holocene CH4 change on simulated temperature. We plan to do this based on already existing idealized KCM experiments. Such experiments were carried out with a 1% p.a.  $CO_2$  increase, and with/without a 1%/2% p.a. CH4 increase (Biastoch et al., 2011, supplementary material Fig. S3). From the 1%p.a. CH4 until +25% CH4 increase experiment (from around 800 to 1000 ppbv) we diagnose a temperature increase of 0.2C over 100 years, from the 2% p.a. until +225% CH4 (800 - 2600 ppbv) a temperature increase of 0.75C. It is difficult to estimate, however, what the climate sensitivity on longer time scales would be without actually performing the experiments, but a contribution on the order of 0.1C to the simulated Holocene global mean SST variation from the prescribed CH4 seems a reasonably conservative estimate.

We will revise Fig. 1a, Sec.2.2.1, Sec. 3.1.1 and the Discussion to include time series of atmospheric methane and the potential impacts on SST-evolution.

**3) Are planetary and cloud albedo included in the radiation calculation?**

Both planetary and cloud albedo are included in the radiation scheme of ECHAM5 (sections 6.1.1 and 11.3.2 in the Technical Report (Roeckner et al., 2003). A more detailed analysis of the atmospheric variations in the KCM-experiments is planned to be published in a separate manuscript in a more climate focussed journal.

**4) Comparison with proxy reconstructions would be nice to see (AMOC, $\delta^{15}N$ )**

A more detailed analysis of the physical ocean variations, possibly including a more in
depth comparison of the model results for the North Atlantic and the tropical Pacific with proxy data would likewise be the topic of a separate study. But we will try to address the issue within our current limitations.

**4.a)** AMOC (Atlantic Meridional Overturning Circulation, data from Hillaire-Marcel et al., 2001, Hoogakker et al., 2011, 2015, Thornalley et al., 2013)

Here we would like to refer to the work of Blaschek et al. (2015), who describe a set of experiments with the earth system model of intermediate complexity "LOVECLIM" and compare their results with various proxies for AMOC (including those investigated by Hoogakker et al. (2011), see Table 2 in Blaschek et al. (2015)). Since the temporal evolution of AMOC in our experiment KCM-HOL is similar to that of experiment 'OG' (Orbital and Greenhouse gases as forcing) in Blaschek et al. (2015), we can assume that their findings are also valid for our experiments: Additional forcing with the 8.2 kyr BP fresh water pulse and also ice sheet topography changes seems to be required to simulate the weak early Holocene AMOC derived from proxies. As those forcings are not included in our model experiment, there is a further reason to discuss our experiments as a sensitivity experiment to orbital and GHG forcing.

**4.b)** Oxygen minimum zones/  $\delta^{15}N$  records in the Arabian Sea and the Eastern Equatorial Pacific

Arabian Sea (AS):

Here we would like to refer to the study of Gaye et al. (2017) in which an earlier version of the accelerated experiment BGC-HOLx10 was compared to observation based estimates of Holocene OMZ evolution in the Arabian Sea. Based on  $\delta^{15}N$  records,
Gaye et al. (2017) find that the AS OMZ has intensified since the last LGM, and that most of this increase occured throughout the Holocene (their Fig. 6). The model results presented here show a more modest increase in AS OMZ-volume than the one in Gaye et al. (2017) even in the non-accelerated experiment, and a rather constant OMZ volume in the accelerated experiment.

Eastern Equatorial Pacific (EEP):

Comparing our results with the proxy-derived estimates of OMZ intensity in the Eastern Tropical South Pacific (Salvatteci et al., 2016), we find an indication of stronger denitrification towards the late Holocene in the proxy data. This would likely support our result of an expanding EEP OMZ, but this could also be a more local decrease in  $O_2$ -concentration, rather than a general expansion of the OMZ. Moreover, the proxy-data for the early Holocene show a decrease in  $\delta^{15}N$ , indicating increasing oxygen concentrations, which is, however, not simulated by our model.

In summary, as we do not simulate nitrogen isotopes (or other proxies), a direct comparison between proxies and model results is somewhat limited. To address Rev#2's comment, we will add the above attempts to the Discussion and point out the limitated nature of that comparison. A more comprehensive comparison of proxy data and our model results has been planned for the future.

**Response to minor comments**

p15, In 6: should indeed be -0.4 GtC/yr
p16, In 20: double 'relevance' will be removed

p19, In 24: 'seaice' will be corrected to 'sea-ice'

p22, In 28: 'effect' will be corrected to 'affect'

p24, In 23: 'pysical' will be corrected to 'physical'

p24, ln 23: 'extrema' will be corrected to 'extremes'

**References**

Augustin, L., Barbante, C., Barnes, P. R. F., Barnola, J.-M., Bigler, M., Castellano, E., Cattani, O., Chappellaz, J. A., Dahl-Jensen, D., Delmonte, B., Dreyfus, G., Durand, G., Falourd, S., Fischer, H., Flückiger, J., Hansson, M. E., Huybrechts, P., Jugie, G., Johnsen, S. J., Jouzel, J., Kaufmann, P. R., Kipfstuhl, S., Lambert, F., Lipenkov, V. Y., Littot, G. C., Longinelli, A., Lorrain, R. D., Maggi, V., Masson-Delmotte, V., Miller, H., Mulvaney, R., Oerlemans, J., Oerter, H., Orombelli, G., Parrenin, F., Peel, D. A., Petit, J.-R., Raynaud, D., Ritz, C., Ruth, U., Schwander, J., Siegenthaler, U., Souchez, R., Stauffer, B., Steffensen, J. P., Stenni, B., Stocker, T. F., Tabacco, I., Udisti, R., van de Wal, R. S. W., van den Broeke, M. R., Wilhelms, F., Winther, J.-G., Wolff, E. W., and Zucchelli, M.: Data from the EPICA Dome C ice core EDC, doi:10.1594/PANGAEA.728149, https://doi.org/10.1594/PANGAEA.728149, supplement to: Augustin, L et al. (2004): Eight glacial cycles from an Antarctic ice core. Nature, 429(6992), 623-628, https://doi.org/10.1038/nature02599, 2004.

Biastoch, A., Treude, T., Rüpke, L. H., Riebesell, U., Roth, C., Burwicz, E. B., Park, W., Latif, M., Böning, C. W., Madec, G., and Wallmann, K.: Rising Arctic Ocean temperatures cause
gas hydrate destabilization and ocean acidification, Geophys. Res. Lett., 38, doi:10.1029/ 20011GL047222, 2011.

- Blaschek, M., Renssen, H., Kissel, C., and Thornalley, D.: Holocene North Atlantic Overturning in an atmosphere-ocean-sea ice model compared to proxy-based reconstructions, Paleoceanography, 30, 1503–1524, doi:10.1002/2015PA002828, http://dx.doi.org/10.1002/ 2015PA002828, 2015PA002828, 2015.
- Gaye, B., Böll, A., Segschneider, J., Burdanowitz, N., Emeis, K.-C., Ramaswamy, V., Lahajnar, N., Lückge, A., and Rixen, T.: Glacial-Interglacial changes and Holocene variations in Arabian Sea denitrification, Biogeosciences, 15, 507–527, doi:10.5194/bg-15-507-2018, https://www.biogeosciences.net/15/507/2018/, 2017.
- Hoogakker, B. A. A., Chapman, M. R., McCave, I. N., Hillaire-Marcel, C., Ellison, C. R. W., Hall, I. R., and Telford, R. J.: Dynamics of North Atlantic Deep Water masses during the Holocene, Paleoceanography, 26, doi:10.1029/2011PA002155, http://dx.doi.org/10.1029/2011PA002155, pA4214, 2011.
- Liu, Z., Zhu, J., Rosenthal, Y., Zhang, X., Otto-Bliesner, B. L., Timmermann, A., Smith, R. S., Lohmann, G., Zheng, W., and Elison Timm, O.: The Holocene temperature conundrum, Proceedings of the National Academy of Sciences, doi:10.1073/pnas.1407229111, http://www. pnas.org/content/early/2014/08/07/1407229111, 2014.
- Roeckner, E., Bäuml, G., Bonaventura, L., Brokopf, R., and Esch, M.: The atmospheric general circulation model ECHAM5. Part I: model description, Report 349, Max Planck Institute for Meteorology, 2003.
- Salvatteci, R., Gutierrez, D., Sifeddine, A., Ortlieb, L., Druffel, E., Boussafir, M., and Schneider, R.: Centennial to millennial-scale changes in oxygenation and productivity in the Eastern Tropical South Pacific during the last 25 000 years, Quaternary Science Reviews, 131, 102–117, 2016.

BGD
Fig. 1. Revised Fig. 1a

---

## Author Response (AR1)

**BGD-2017-554, Segschneider et al.: Response to Rev#1**

**Climate and marine biogeochemistry during the Holocene from transient model simulations**
We wish to thank Rev#1 for his many thoughtful comments and in particular for pointing us to the work of Liu et al. (2014) on the Holocene climate conundrum. We performed additional analyses to address the reviewer's comments where possible and also extended the control experiments KCM-CTL and BGC-CTL further. Comments are adressed point-by-point in the following. In addition we divided the Discussion into subsections for better readability and moved the 'North Atlantic' section from Results to Discussion (Sec. 4.3).

**1) Role of model drift**

In deed the control run of 'only' 2000 years turned out to be non-satisfactory for the nonaccelerated experiments, and Rev#1 is right in that we could not exclude that at least part of the signals we discuss are remaining model drift or internal model variability. 2000 years seemed quite long with regard to the accelerated 950 year-long experiments. It was then unexpected that there was still some model drift (and/or internal variability) that amounted, for some parameters, to variability of similar magnitude as the variations of the non-accelerated transient experiments.

The basis for the KCM experiments is a 1,000 year KCM experiment with 9.5k orbital parameters, 286.6 ppm CO2, and 805 ppbv CH4 concentration (with a final global average SST of 15.8°C), followed by a spin up for a further 1,000 years with 9.5kyr BP orbital and GHG forcing. From this state the KCM-CTL and KCM-HOL experiments were started. Apparently the 2x 1,000 year spinup time was still not long enough, and a model drift remained in KCM-CTL that was significant given the small temperature changes that occured during the Holocene. We have become aware of this problem during the analyses of experiment KCM-HOL, and extended the control runs (KCM-CTL and BGC-CTL) since, but they are not finished yet and will

still need up to 2 more month to do so (currently at 8100 years, 1.6 kyr BP).

This has an impact of the interpretation of the SST (revised Fig. 2a, see also response to FC 7), and to some extent on the biogeochemical variables including the EEP OMZ (revised Fig. 13). We now mention this in the abstract (p3, ln 8-12), Section 2.3 and 2.4 (description of KCM/BGC experiments), and more generally in the Results and Discussion sections.
We also note that for all the paleo experiments in the literature, there are no(!) control experiments to be found for transient and time-slice experiments (Fischer and Jungclaus, 2011; Varma et al., 2016; Liu et al., 2014; Bopp et al., 2017). So this is a more general problem in the paleo climate modelling community, probably based on an underestimation of model drift and internal model variability compared to the relatively weak forced Holocene variability on long time scales.

**2) Off-line biogeochemistry**

Convection and diffusion are simulated as in the online version, based on the stored mixing parameters from OPA9 as part of KCM. So there might be some differences from the monthly time-averaging of the online output and the non-linearity of mixing, but we have not investigated this systematically for our model configuration as there is no online version of KCM-PISCES available.

For the same reason, we can not make a statement about potential differences of water mass ages in the on- and off-line versions of PISCES.

**3) Vertical resolution of the ocean**

There are 20 layers below 100m depth, and layer thickness near the bottom is about 500m. We have no indication that this causes problems for the simulation of deep Pacific  $O_2$  within the general limitations due to the coarse model resolution.

We now describe the vertical resolution of the ocean model in the KCM model description.
**Discussion** Paper
**4) Missing processes, potential impact on results should be discussed**

- sediment is not included in model setup
- no changes in weathering
- no coral reefs (Vecsei and Berger, 2004)

In setting up our experiments, we followed the standard PMIP protocol. So while the above processes are not included in the model setup, their combined effects on GHG climate forcing must be represented by the reconstructed PMIP GHG forcing from ice cores. With regard to the evolution of atmospheric  $CO_2$ , we now point to the study of Brovkin et al. (2016), were all these processes are described in fair detail and nicely summarized in their Fig. 8. We note, however, that changes in alkalinity due to changes in planktonic calcification as they occur in our model, are not included in the study of Brovkin et al. (2016).

- how is freshwater pulse of 8.2 kyr BP represented in the forcing?

The freshwater pulse at 8.2 kyr BP is not included in the forcing, and hence we can not expect to find changes in AMOC related to it. This is now picked up in the description and discussion of AMOC (Sec. 3.1.2 and Discussion 4.2).

- role of changes in the land biopshere carbon inventory

The land biosphere is not included in the model, but we use reconstructed  $pCO_2$  and other greenhouse gases, which should include any contribution of the land biosphere to atmospheric

GHG-concentrations. So what remains neglected is albedo changes from changes in vegetation, but we can not quantify the potential impact this would have on the simulated climate.

- potential influence from volcanic eruptions?

Including volcanic aerosol forcing would likely have a cooling effect during the first 1-2 years following the volcanic eruptions, but this is difficult to quantify for KCM without performing the experiment. For the same reason, also any integrated effect of volcanic eruptions on long term evolution of temperature is difficult to establish, but would likely to be small based on the coupled climate model experiments for the last millennium (Brovkin et al., 2010) that indicate a  $-0.8^{\circ}$  cooling for the 1258 eruption (the largest eruption during the simulated period), but within a decade surface air temperature fluctuations are within the background range (their Fig. 1). Any effect on atmospheric pCO2 related to post-glacial increased volcanic activity and additional outgassing of 1,000-5,000 GtCO2 between 12 kyr and 7 kyr BP (Huybers and Langmuir, 2009) will be included in the prescribed PMIP GHG-forcing (which shows decreasing pCO2 during the early Holocene).

In summary, while these are all interesting points, we do not want to hypothesize about the potential effects of the omitted forcings and model components.

Also in response to Rev#2, we now describe our model experiments as sensitivity experiments to the PMIP orbital and GHG forcings with likely deviations from the Holocene climate variations that are caused by other forcings and biogeochemical processes that are not included in our model setup. We explicitly mention the neglected components and forcings (Section 2.1.1), and indicate where this might have a direct impact on our results. We have also extended the more general discussion of differences between proxy and model-based evolution of Holocene climate in the revised section 4.1.

4

**5) Interpretation of $\mathbf{CO}_2$ fluxes in the light of prescribed $\mathbf{pCO}_2$**

We think that we were quite careful in our wording as to how much the diagnosed  $CO_2$ -fluxes can be meaningful for a quantification of a potential contribution of the ocean to atmospheric pCO2, however, seemingly not careful enough.
We deleted Fig. 7 and included the time-integrated ocean-atmosphere carbon flux in a revised Fig. 6a as suggested. We also revised the description of the time-integrated carbon flux and its discussion (Sections 3.2.1 and 4.5).

**6) Attribution of changes of $O_2$ and other tracers to underlying processes**

- stratification changes and impact on  $\ensuremath{O_2}$

In contrast to studies based on global warming and  $O_2$  (Cocco et al., 2013; Bopp et al., 2013) and marine biological production (Steinacher et al., 2010), that showed an impact of stratification changes on  $O_2$  fields and marine biological production, here we simulate much smaller temperature variations, but much longer periods. Global and annual mean of the MLD in KCM-HOL reveal little temporal variability: MLD is around 48m at 9.5kyr BP, and starts to decrease after 5.5 kyr BP to around 47m at 0 kyr BP. We, therefore, state that stratification changes play only a minor role for  $O_2$  changes during the Holocene (revised Sec. 4.4).

- $O_2$  saturation and AOU

We computed also the fields of  $O_2$ -saturation and AOU for experiment BGC-HOL (new Fig. 13) and included the description and discussion in Sections 3.2.5 and 4.4.

**7) export production wrong metric to judge efficiency of the biological pump**

With due respect, on this point we disagree with Rev#1. A large number of studies describe the biological carbon pump, also called soft tissue pump as driven by the export production (e.g. Six and Maier-Reimer, 1996; Sigman and Hain, 2012; Ducklow et al., 2001), and the whole JGOFS project was based on this principle. See also https://www.us-ocb.org/biological-pump/ for a very condensed description. Evidently the export production is related to the uptake of nutrients in the euphotic zone, but as not all of the nutrients that are taken up in the euphotic zone are exported to depth because of remineralization and grazing in the euphotic layer itself, we emphasize that export production is a correct metric for the strength of the biological carbon pump.
We did not change our statement regarding the biological pump (now in Section 4.5), but included references to the biological pump and its role for atmosphere-ocean carbon fluxes.

**Response to further Comments (FC)**

**FC 1) Other transient simulations not mentioned (p4, 2nd para)**

We referenced some exemplary transient simulations rather than trying to give a complete overview (again, our focus being more on the marine biogeochemistry), but to address this point we included in the Introduction references to Brovkin et al. (2016) as an example of EMIC experiments, and to Liu et al. (2014) as an example for the 21ka experiment, and to Fischer and Jungclaus (2011) as a further non-accelerated coupled model experiment from mid-to-late Holocene.

**FC 2) Are changes in ice albedo taken into account? (p9)**

Changes in sea-ice cover are simulated by LIM, the sea-ice component of NEMO, but that is

probably not meant here. We now mention explicitly in Section 2.2 that we do not take into account solar TSI and volcanic forcing, nor changes in the continental ice sheets (neither topography nor albedo) and also no fresh water pulses.

**FC 3) Forcing data GHG**

We admit somewhat shamefacedly that due to a misunderstanding we stated that we force KCM-HOL with only  $CO_2$  as greenhouse gas. However, the experiments were also forced with transient  $CH_4$  and  $N_2O$  from the PMIP data base. This mistake in the description of the forcing, however, does not change the findings of our study.

We rewrote section 2.2.1 accordingly and revised Fig. 1a to include the time series of  $CH_4$  and  $N_2O$  (see above). We now also provide the internet address from where we obtained the data, and added the reference to Augustin et al. (2004) describing the EPICA data.

**FC 4) 1st para would better fit in methods sections (p12, ln 2)**

We moved the paragraph to a new section 2.3 'Processing of model output' in the Methods section.

**FC 5) delete 'again' p12, ln 21**

has been deleted

FC 6) drift is not 'modest', please delete modest (p12, ln 25)

**FC 7) It seems the whole [SST] signal may be explained by drift? (p13, ln1-2)**

See also response to main comment **1**). We have extended experiment KCM-CTL in the meantime by a further 5800 years (leading up to 1.6 kyr BP). We now state that parts of the initial SST decrease in KCM-HOL can indeed be explained by the drift (the decrease is stronger in KCM-HOL), while the following SST increase is damped by the model drift (which becomes smaller after 6 kyr BP). As a result, the initial SST decrease would be weaker in a drift-free setup, while the following SST increase would be stronger. It would of course be ideal to run the extended control experiment until 0k, but that would potentially delay publication by several months (a minimum of 1.5 months, from past experience more likely 2-4 months).

We revised Section 3.1.1 accordingly, and ammended Fig. 2a to include the extended control run up to 1.6 kyr BP and to include the SST drift averaged over the first 500 years (grey bar), which is very small and led us to assume the spin-up/control experiment was already in balance.

**FC 8) Indo Pacific overturning should be discussed also (p13, sec. 3.1.2)**

In response to FC8 we also analysed the Indo Pacifc overturning. We included a time series of Pacific maximum meridional streamfunction between 3000 and 5000m depth at the equator in a revised Fig. 3 and added the description and discussion on deep Pacific northward inflow in Section 3.1.2 and 4.2.

**FC 11) Authors should say sth. on SST evolution at different seasons (p20, Discussion)**

While we felt that this goes a bit beyond the scope of our manuscript (and possibly the focus of Biogeosciences), we analysed the summer (JJA) and winter (DJF) SST separately. We added to Section 4.1 that the simulated SST evolution in KCM-HOL is seemingly not very sensitive to the choice of season. We further point to the study of Liu et al., 2014 for a more detailed analysis of the general proxy-model mismatch for MH temperatures and its seasonal dependency (Section 4.1).

**FC 12) How was BGC-CTL extendend (Fig. 5)**

BGC-CTL was extended by forcing PISCES for another -repeating - cycle of the 2000 yrs available from KCM-CTL. This extension has been replaced by the now extended control runs in revised Figures 2-12,14,15.

Further errors found by the authors

OMZ-volume for the Arabian Sea erroneously showed values for a small region off Peru, for which the main author had made some quick analyses for model-proxy comparison and then did not change the script back to the Arabian Sea lon/lat bounds. The Arabian Sea time series is now very similar to the one published in Gaye et al. (2017). We apologize for this mistake.

Typo in Ref of (Leduc et al., 2010): Ma/Ca was corrected to Mg/Ca

E.g. misplaced p23 ln 3

**Response to minor comments**

- p15, ln 6: corrected to -0.4 GtC/yr.
- p16, ln 20: double 'relevance' has been removed
- p19, ln 24: 'seaice' was corrected to 'sea-ice'
- p22, ln 28: 'effect' was corrected to 'affect'
- p24, ln 23: 'pysical' was corrected to 'physical'
- p24, ln 23: 'extrema' was corrected to 'extremes'

**3.2.6 North Atlantic**

5

Our original intention in examining the North Atlantic more closely was to investigate whether the changes in the OMZs could be traced back to the deep water source regions. It turned out,

- however, that significant changes occurred in the North Atlantic, that justify further analysis. In section 3.1.3 we showed a sudden drop in the water mass age in the deep North Atlantic (Fig. 4a) that can be traced back to a westward shift in the deep water formation areas south of Iceland, and a northward shift north of Iceland, as indicated by the difference in the annual maximum of
- the mixed layer depth in the North Atlantic (Fig. 4b). In addition to the shift of location, also an increase of the mixed layer depth of up to 3000 m occurs in the more southwestern part of the Nordic Seas. This is accompanied by a change in SST at 60N around 6.8 kyr BP from negative to positive anomalies (Fig. 2b). and an increase in export production (Fig.10b).
**Discussion** Paper

- An SST time series at 53N, 30W shows rapid decrease in SST, a time series at 62N 30W a slight SST increase, and both time series have reduced variability after 7 kyr BP (figure not shown). Fig. 1b reveals a shift from negative anomalies of annual mean SWR at the ocean/seaice surface just north of 60N to positive anomalies in a narrow band south of 60N, and a negative anomaly north of 75N. In particular the positive anomaly south of 60N 
[revised manuscript text omitted]

**Fig. A.1.** Simulated and observation based observation-based profiles of average O2-concentration in  $\mu$ mol l-1 in the three major oxygen minimum zones in the world ocean for (**a**) the eastern equatorial Pacific, (**b**) the tropical South Atlantic, and (**c**) the Arabian Seabased. Based on observations (WOA2013, solid) (WOA2013, Garcia et al., 2013, solid), and from experiments BGC-CTL (dotted) and BGC-HOL (dashed) averaged over the last 200 years of BGC HOL (dashed).

**Fig. A.2.** Simulated and observation based observation-based (WOA2013, Garcia et al., 2013) volume of water masses with oxygen concentration below the threshold value indicated on the x-axis for (a) the world ocean from the surface to the seafloor in  $10^{15}$  m3, and (b) the eastern equatorial Pacific for 0 - 1000 m and 0 - 5000 m in  $10^{14}$  m3. See legends for explanations of symbols.

**Fig. A.3.** Time series of ocean atmosphere (a) SST and (b) atmosphere-ocean carbon flux (negative upward) for experiment BGC-HOL (black dots: annual mean values, solid black line: 4th order 4th-order polynomial fit), and annual minimum and maximum (dashed black lines, 4th order 4th-order polynomial fits), indicating the range of the annual cycle.

---

## Author Response (AR2)

**BG-2017-554, Segschneider et al.: Response to Editor**

**Climate and marine biogeochemistry during the Holocene from transient model simulations**

Dear Christoph Heinze,

thank you for clarifying the issue regarding strength and efficiency of the biological pump. Following your suggestion we have computed a time series of the efficiency of the biological pump following Sarmiento and Gruber (2006) (see below, not included in revised ms.) and included a paragraph on the temporal evolution in the discussion.

[Figure]

**Fig. 1.** Efficiency of the biolgical pump based on eq. 4.1.1 in Sarmiento and Gruber (2006) and global mean NO3 from 0-100m (surface) and 100-200 m (deep) as in Fig. 4.1.7 in Sarmiento and Gruber (2006). X-axis in years/10, 950 = 0 kyr BP.

[revised manuscript text omitted]

---

## Author Response (AR3)

**Subject:** response 1 Re: FW: bg-2017-554 (author) - manuscript accepted for final publication
**From:** Christoph Heinze <Christoph.Heinze@uib.no>
**Date:** 08/05/18 09:35
**To:** Anna Wenzel <anna.wenzel@copernicus.org>
**CC:** joachim.segschneider@ifg.uni-kiel.de

```
Dear All,
I agree to include this update in the final version for upload, Joachim.
It would be good, however, to document this in the email exchange stored in the BG
archive for the manuscript ("MS records"),
so that in case of questions we can reconstruct what happened. Can you arrange for this
Anna?
Many thanks,
Christoph Heinze
* * *
Christoph Heinze
prof. in chemical oceanography,
University of Bergen, Geophysical Institute,
Bjerknes Centre for Climate Research,
and Uni Research Climate
Allégaten 70, N-5007 Bergen, Norway
Mobile phone work: +47 975 57 119
Email: christoph.heinze@uib.no
CURRENTLY ON SABBATICAL AT GEOMAR, Kiel
land line +49 431 600 4212

On 2018-05-08 09:22, Anna Wenzel wrote:
 Dear Christoph,

 The authors asked me to forward the attached message to you.

 Kind regards,

 Anna
* * *
 Copernicus Publications
 The Innovative Open-Access Publisher

 Anna Wenzel
 Editorial Support

 Copernicus GmbH
 Bahnhofsallee 1e
 37081 Göttingen
 Germany

 Phone: +49 551 90 03 39 43
 Fax: +49 551 90 03 39 90 43

 http://www.copernicus.org
 @copernicus_org
* * *
 Copernicus Gesellschaft mbH
 USt-IdNr.: DE216566440
 Based in Göttingen, Germany
```

Registered in HRB 131 298
County Court Göttingen
Managing Director Thies Martin Rasmussen
* * *
From: j.segschneider [mailto:joachim.segschneider@ifg.uni-kiel.de]
Sent: 07 May 2018 16:36
To: Copernicus Publications Editorial Support
Cc: birgit.schneider@ifg.uni-kiel.de
Subject: Re: bg-2017-554 (author) - manuscript accepted for final publication

Dear Natascha Töpfer,

if possible I would appreciate if you could forward the below enquiry
to the editor of
the above paper, unless any formal reasons speak against this.

Best regards, Joachim Segschneider

Dear Christoph Heinze,

Since the control-experiment KCM/BGC-CTL has completed another 800
years in the meantime,
I would consider it useful to provide the figures with the additional
model years in the publication
of the above manuscript since they provide a more complete view.

I have attached a zip-archive of the updated figures (.jpg).

In addition this would require updated CTL-related numbers in Tab.1
(runtime) and in a few places in the results
section (state of -CTL at end of CTL experiments) but would not change
any of the findings of the paper.

Please let me know if I can upload the updated figures, or if you
would prefer the formally accepted
versions.

Waiting for the CTL-experiment to complete the full 9,500 years would
still require 1-2 months.

In the hope of a positive reply and best regards

Joachim Segschneider

On 02/05/18 13:54,
editorial@copernicus.org<mailto:editorial@copernicus.org> wrote:

Dear Joachim Segschneider,

We are pleased to inform you that your following manuscript was
accepted for final publication in BG:

Title: Climate and marine biogeochemistry during the Holocene from transient model simulations

Author(s): Joachim Segschneider et al.

MS No.: bg-2017-554

MS Type: Research article

Iteration: Minor Revision

Special Issue: Progress in quantifying ocean biogeochemistry - in honour of Ernst Maier-Reimer

Presently, your manuscript is being transferred to the Copernicus Publications Production Office for typesetting and publication. To proceed, please upload all files that are required for production no later than 12 May 2018 at https://editor.copernicus.org/BG/production_file_upload/bg-2017-554. For further information on files and formats we kindly refer you to the submission guidelines: https://www.biogeosciences.net/for_authors/submit_your_manuscript.html

In your manuscript, please use full first names for all authors. Although references are still based on initials, we will use full first names on the title page of your paper.

To log in, please use your Copernicus Office user ID 27271.

You are invited to monitor the processing of your manuscript via your MS Overview: https://editor.copernicus.org/BG/my_manuscript_overview

In case any questions arise, please contact me.

Kind regards,

Natascha Töpfer

Copernicus Publications

Editorial Support

editorial@copernicus.org<mailto:editorial@copernicus.org>

on behalf of the BG Editorial Board

--

Joachim Segschneider

Christian-Albrechts-Universität zu Kiel

Ludewig-Meyn-Str. 10

Room 113

D-24118 Kiel

ph. ++49 431 880 2861

email: joachim.segschneider@ifg.uni-kiel.de<mailto:joachim.segschneider@ifg.uni-kiel.de>

**Subject:** response 2 Re: FW: bg-2017-554 (author) - manuscript accepted for final publication
**From:** Christoph Heinze <Christoph.Heinze@uib.no>
**Date:** 08/05/18 09:37
**To:** Anna Wenzel <anna.wenzel@copernicus.org>
**CC:** joachim.segschneider@ifg.uni-kiel.de

Dear All,
I agree to ALSO include this ADDITIONAL update in the final version for upload,
Joachim. NO PROBLEM.
It would be good, however, to document this in the email exchange stored in the BG
archive for the manuscript ("MS records"),
so that in case of questions we can reconstruct what happened. Can you arrange for this
Anna?
Many thanks,
Christoph Heinze
* * *
Christoph Heinze
prof. in chemical oceanography,
University of Bergen, Geophysical Institute,
Bjerknes Centre for Climate Research,
and Uni Research Climate
Allégaten 70, N-5007 Bergen, Norway
Mobile phone work: +47 975 57 119
Email: christoph.heinze@uib.no
CURRENTLY ON SABBATICAL AT GEOMAR, Kiel
land line +49 431 600 4212

On 2018-05-08 09:22, Anna Wenzel wrote:

> Dear Christoph,
>
> Please find attached the follow-up email.
>
> Kind regards,
>
> Anna
>
> **************************************************
> Copernicus Publications
> The Innovative Open-Access Publisher
>
> Anna Wenzel
> Editorial Support
>
> Copernicus GmbH
> Bahnhofsallee 1e
> 37081 Göttingen
> Germany
>
> Phone: +49 551 90 03 39 43
> Fax: +49 551 90 03 39 90 43
>
> http://www.copernicus.org
> @copernicus_org
> **************************************************
> Copernicus Gesellschaft mbH

USt-IdNr.: DE216566440
Based in Göttingen, Germany
Registered in HRB 131 298
County Court Göttingen
Managing Director Thies Martin Rasmussen
* * *
From: j.segschneider [mailto:joachim.segschneider@ifg.uni-kiel.de]
Sent: 07 May 2018 17:01
To: Copernicus Publications Editorial Support
Subject: Re: bg-2017-554 (author) - manuscript accepted for final publication

Dear Natascha Töpfer, Dear Christoph Heinze,

maybe the attached text that highlights the required changes is helpful in
making the decision with regard to including updated figures.

Changes can be found on p5, p7, p8, and Table 1 (p19).

Sorry for not supplying this with my earlier email.

Best regards, Joachim Segschneider

On 02/05/18 13:54,
editorial@copernicus.org<mailto:editorial@copernicus.org> wrote:

Dear Joachim Segschneider,

We are pleased to inform you that your following manuscript was
accepted for final publication in BG:

Title: Climate and marine biogeochemistry during the Holocene from
transient model simulations

Author(s): Joachim Segschneider et al.

MS No.: bg-2017-554

MS Type: Research article

Iteration: Minor Revision

Special Issue: Progress in quantifying ocean biogeochemistry - in
honour of Ernst Maier-Reimer

Presently, your manuscript is being transferred to the Copernicus
Publications Production Office for typesetting and publication. To
proceed, please upload all files that are required for production no
later than 12 May 2018 at
https://editor.copernicus.org/BG/production_file_upload/bg-2017-554.
For further information on files and formats we kindly refer you to
the submission guidelines:
https://www.biogeosciences.net/for_authors/submit_your_manuscript.html

In your manuscript, please use full first names for all authors.
Although references are still based on initials, we will use full
first names on the title page of your paper.

To log in, please use your Copernicus Office user ID 27271.

You are invited to monitor the processing of your manuscript via your
MS Overview: https://editor.copernicus.org/BG/my_manuscript_overview

In case any questions arise, please contact me.

Kind regards,

Natascha Töpfer

Copernicus Publications

Editorial Support

editorial@copernicus.org<mailto:editorial@copernicus.org>

on behalf of the BG Editorial Board

--

Joachim Segschneider

Christian-Albrechts-Universität zu Kiel

Ludewig-Meyn-Str. 10

Room 113

D-24118 Kiel

ph. ++49 431 880 2861

email:
joachim.segschneider@ifg.uni-kiel.de<mailto:joachim.segschneider@ifg.uni-kiel.de>

[revised manuscript text omitted]